# THE OPTIMALITY OF KERNEL CLASSIFIERS IN SOBOLEV SPACE

**Jianfa Lai**[*]
Tsinghua University, Beijing, China
jianfalai@mail.tsinghua.edu.cn

**Zhifan Li**[*]
Beijing Institute of Mathematical Sciences
and Applications, Beijing, China
zhifanli@bimsa.cn

**Dongming Huang**
National University of Singapore, Singapore
stahd@nus.edu.sg

**Qian Lin**[†]
Tsinghua University, Beijing, China
qianlin@tsinghua.edu.cn

## ABSTRACT

Kernel methods are widely used in machine learning, especially for classification problems. However, the theoretical analysis of kernel classification is still limited. This paper investigates the statistical performances of kernel classifiers. With some mild assumptions on the conditional probability $\eta(x) = \mathbb{P}(Y = 1 \mid X = x)$, we derive an upper bound on the classification excess risk of a kernel classifier using recent advances in the theory of kernel regression. We also obtain a minimax lower bound for Sobolev spaces, which shows the optimality of the proposed classifier. Our theoretical results can be extended to the generalization error of overparameterized neural network classifiers. To make our theoretical results more applicable in realistic settings, we also propose a simple method to estimate the interpolation smoothness of $2\eta(x) - 1$ and apply the method to real datasets.

## 1 INTRODUCTION

In this paper, we study the problem of binary classification in a reproducing kernel Hilbert space (RKHS). Suppose $n$ i.i.d samples $\{(X_i, Y_i) \in \mathcal{X} \times \{-1, 1\}\}$ are drawn from a joint distribution $(X, Y) \sim \rho$, where the conditional probability of the response variable $Y$ given the predictor variable $X = x$ is denoted by $\eta(x) = \mathbb{P}(Y = 1 | X = x)$. We aim to find a classifier function $f(x) : \mathcal{X} \to [-1, 1]$ that minimizes the classification risk, defined as:

$$\mathcal{L}(\hat{f}) := \mathbb{P}_{(X,Y)\sim\rho} \left[ \text{sign}(\hat{f}(X)) \neq Y \right].$$

The minimal classification risk is achieved by the Bayes classifier function corresponding to $\rho$, which is defined as $f_\rho^*(x) = 2\eta(x) - 1$. Our main focus is on analyzing the convergence rate of the classification excess risk, defined as:

$$\mathcal{E}(\hat{f}) = \mathcal{L}(\hat{f}) - \mathcal{L}(f_\rho^*).$$

This paper studies a class of kernel methods called spectral algorithms (which will be defined in Section 2.3) for constructing estimators of $f_\rho^*$. The candidate functions are selected from an RKHS $\mathcal{H}$, which is a separable Hilbert space associated with a kernel function $K$ defined on $\mathcal{X}$ (Smale & Zhou, 2007; Steinwart & Christmann, 2008). Spectral algorithms, as well as kernel methods, are becoming increasingly important in machine learning because both experimental and theoretical results show that overparameterized neural network classifiers exhibit similar behavior to classifiers based on kernel methods Belkin et al. (2018). Therefore, understanding the properties of classification with spectral algorithms can shed light on the generalization of deep learning classifiers.

In kernel methods context, researchers often assume that $f_\rho^* \in \mathcal{H}$, and have obtained the minimax optimality of spectral algorithms (Caponnetto, 2006; Caponnetto & De Vito, 2007). Some

---

[*]These authors contributed equally to this work.
[†]Corresponding author.

researchers have also studied the convergence rate of the generalization error of misspecified spectral algorithms $\left(f_\rho^* \notin \mathcal{H}\right)$, assuming that $f_\rho^*$ falls into the interpolation space $[\mathcal{H}]^s$ with some $s > 0$ (Fischer & Steinwart, 2020; Zhang et al., 2023). In this line of work, researchers consider the embedding index condition which reflects the capability of $\mathcal{H}$ embedding into $L^\infty$ space. Moreover, Zhang et al. (2023) extends the boundedness assumption to the cases where $[\mathcal{H}]^s \cap L^\infty(X, \mu) \subsetneqq [\mathcal{H}]^s$.

Motivated by the aforementioned studies, we adopt similar assumptions in our study of kernel classifiers trained via the gradient flow. We assume that the Bayes classifier $f_\rho^* \in [\mathcal{H}]^s$ satisfies the boundedness condition $f_\rho^* \in [-1, 1]$. We first derive the upper bound of the classification excess risk, showing that the generalization error of the kernel classifier is highly related to the interpolation smoothness $s$. To clarify the minimax optimality of kernel classification, we then obtain the minimax lower bound for classification in Sobolev RKHS, which is a novel result in the literature. Our technique is motivated by the connection between kernel estimation and infinite-width neural networks, and our framework can be applied to neural network supervised learning. Furthermore, we provide a method to estimate the interpolation space smoothness parameter $s$ and also present some numerical results for neural network classification problems through simulation studies and real data analysis.

## 1.1 OUR CONTRIBUTION

In this paper, we study the generalization error of kernel classifiers. We show that

$i$) We show the generalization error of the gradient flow kernel classifier is bounded by $O(n^{-s\beta/(2s\beta+2)})$ provided that the Bayes classifier $f_\rho^* \in [\mathcal{H}]^s$, where $\beta$ is the eigenvalue decay rate (EDR) of the kernel. This result is not only applicable to the Sobolev RKHS $\mathcal{H}$ but also to any RKHS with the embedding index $\alpha_0 = 1/\beta$, such as the RKHS with dot-product kernels and the RKHS with shift-invariant periodic kernels.

$ii$) We establish a minimax lower bound on the classification excess risk in the interpolation space of Sobolev RKHS. Combined with the results in $i$), the convergence rate of the kernel classifier is minimax optimal in Sobolev space. Before our work, Yang (1999) illustrated a similar result of the minimax lower bound for Besov spaces. However, the result has only been proved for $d = 1$ by Kerkyacharian & Picard (1992) and the case for $d > 1$ remains unresolved.

$iii$) To make our theoretical results more applicable in realistic settings, we propose a simple method to estimate the interpolation smoothness $s$. We apply this method to estimate the relative smoothness of various real datasets with respect to the neural tangent kernels, where the results are in line with our understanding of these real datasets.

## 1.2 RELATED WORKS

We study the classification rules derived from a class of real-valued functions in a reproducing kernel Hilbert space (RKHS), which are used in kernel methods such as Support Vector Machines (SVM) (Steinwart & Christmann, 2008). Most of the existing works consider hinge loss as the loss function, i.e. Wahba (2002); Steinwart & Scovel (2007); Bartlett & Wegkamp (2008); Blanchard et al. (2008) etc. Another kernel method, kernel ridge regression, also known as least-square SVM (Steinwart & Christmann, 2008), is investigated by some researchers (Xiang & Zhou, 2009; Rifkin et al., 2003). Recently, some works have combined the least square loss classification with neural networks (Demirkaya et al., 2020; Hu et al., 2021).

We choose kernel methods because it allows us to use the integral operator tool for analysis (De Vito et al., 2005; Caponnetto & De Vito, 2007; Fischer & Steinwart, 2020; Zhang et al., 2023), while previous SVM works tend to use the empirical process technique (Steinwart & Scovel, 2007). Moreover, we can easily extend the $\mathcal{H}$ to the misspecified model case $[\mathcal{H}]^s$ when true model $f_\rho^*$ belongs to a less-smooth interpolation space. Furthermore, we consider more regularization methods, collectively known as spectral algorithms, which were first proposed and studied by Rosasco et al. (2005); Bauer et al. (2007); Caponnetto & De Vito (2007). Zhang et al. (2023) combined these two ideas and obtained minimax optimality for the regression model. We extend their results to the classification problems.

We study the minimax optimality of Sobolev kernel classification, and before our work, the minimax lower bound of classification excess risk for the RKHS class was seldom considered. Loustau (2008; 2009) have discussed Classification problems in Sobolev space, but they did not consider the lower bound of classification risk. Audibert (2004); Audibert & Tsybakov (2007); Massart & Nédélec (2006) provided some minimax lower bound techniques for classification, but how to solve RKHS remains unknown. Sobolev space (see, e.g., Adams & Fournier (2003)) is known as a vector space of functions equipped with a norm that is a combination of $L^2$-norms of the function together with its derivatives up to a given order and can be embedded into Hölder class. Inspired by the minimax lower bound for Hölder class classification in Audibert & Tsybakov (2007), we derive the lower bound for the Sobolev class.

Recently, deep neural networks have gained incredible success in classification tasks from image classification (Krizhevsky et al., 2012; He et al., 2016) to natural language processing (Devlin et al., 2019). Since Jacot et al. (2018) introduced the neural tangent kernel, The gradient flow of the training process can be well approximated by a simpler gradient flow associated with the NTK kernel when the width of neural networks is sufficiently large (Lai et al., 2023; Li et al., 2023a). Therefore, we can analyze the classification risk of neural networks trained by gradient descent.

## 2 PRELIMINARIES

We observe $n$ samples $\{(X_i, Y_i) \in \mathcal{X} \times \{-1, 1\}\}$ where $\mathcal{X} \subset \mathbb{R}^d$ is compact. Let $\rho$ be an unknown probability distribution on $\mathcal{X} \times \{-1, 1\}$ and $\mu$ be the marginal distribution on $\mathcal{X}$. We assume $\mu$ has a uniformly bounded density $0 < \mu_{min} \leq \mu(x) \leq \mu_{max}$ for $x \in \mathcal{X}$. The classification task is to predict the unobserved label $y$ given a new input $x$. The conditional probability is defined as $\eta(x) = \mathbb{P}(Y = 1 | X = x)$. For any classifier $f$, the risk based on the 0-1 loss can be written as

$$\mathcal{L}(f) = \boldsymbol{E}_{(X,Y) \sim \rho} \mathbb{I}\{\text{sign}(f(X)) \neq Y\} = \boldsymbol{E}_X[(1 - \eta(X))\mathbb{I}\{f(X) \geq 0\} + \eta(X)\mathbb{I}\{f(X) < 0\}]. \tag{1}$$

One of the minimizers of the risk has the form $f_\rho^* = 2\eta - 1$. Let $\mathcal{L}^* = \mathcal{L}(f_\rho^*)$. For any classifier $\hat{f}$ learned from data, its accuracy is often characterized by the classification excess risk, which can be formulated as

$$\mathcal{E}(\hat{f}) = \mathcal{L}(\hat{f}) - \mathcal{L}^* = \boldsymbol{E}_X(|f_\rho^*(X)|\mathbb{I}\{\hat{f}(X)f_\rho^*(X) < 0\}). \tag{2}$$

In the rest of this section, we introduce some essential concepts in RKHS and kernel classifiers. In Section 2.1, we review some definitions in the interpolation space of RKHS. The relationship between fractional Sobolev space and Sobolev RKHS is presented in Section 2.2. Section 2.3 presents the explicit formula of the gradient-flow kernel classifier and the corresponding rewritten form through spectral algorithms and filter functions.

### 2.1 INTERPOLATION SPACE OF RKHS

Denote $L^2(\mathcal{X}) := L^2(\mathcal{X}, \mu)$ as the $L^2$ space. Throughout the paper, we denote by $\mathcal{H}$ a separable RKHS on $\mathcal{X}$ with respect to a continuous kernel function $K$. We also assume that $\sup_{x \in \mathcal{X}} K(x, x) \leq \kappa$ for some constant $\kappa$. The celebrated Mercer's theorem shows that there exist non-negative numbers $\lambda_1 \geq \lambda_2 \geq \cdots$ and functions $e_1, e_2, \cdots \in L^2(\mathcal{X})$ such that $\langle e_i, e_j \rangle_{L^2(\mathcal{X})} = \delta_{ij}$ and

$$K_d(x, x') = \sum_{j=1}^{\infty} \lambda_j e_j(x) e_j(x'), \tag{3}$$

where the series on the right hand side converges in $L^2(\mathcal{X})$.

Denote the natural embedding inclusion operator by $S_k : \mathcal{H} \to L^2(\mathcal{X}, \mu)$. Moreover, the adjoint operator $S_k^* : L^2(\mathcal{X}, \mu) \to \mathcal{H}$ is an integral operator, i.e., for $f \in L^2(\mathcal{X}, \mu)$ and $x \in \mathcal{X}$, we have

$$(S_k^* f)(x) = \int_{\mathcal{X}} K(x, x') f(x') d\mu(x').$$

It is well-known that $S_k$ and $S_k^*$ are Hilbert-Schmidt operators (and thus compact) and their HS norms (denoted as $\|\cdot\|_2$) satisfy that

$$\|S_k^*\|_2 = \|S_k\|_2 = \|K\|_{L^2(\mathcal{X}, \mu)} := \left(\int_{\mathcal{X}} K(x, x) d\mu(x)\right)^{1/2} \leq \kappa.$$

Next, we define two integral operators as follows:

$$L := S_k S_k^* : L^2(\mathcal{X}, \mu) \to L^2(\mathcal{X}, \mu), \quad T := S_k^* S_k : \mathcal{H} \to \mathcal{H}.$$

$L$ and $T$ are self-adjoint, positive-definite, and in the trace class (and thus Hilbert-Schmidt and compact). Their trace norms (denoted as $\| \cdot \|_1$) satisfy that $\|L\|_1 = \|T\|_1 = \|S_k\|_2^2 = \|S_k^*\|_2^2$.

For any $s \geq 0$, the fractional power integral operator $L^s : L^2(\mathcal{X}, \mu) \to L^2(\mathcal{X}, \mu)$ and $T^s : \mathcal{H} \to \mathcal{H}$ are defined as

$$L^s(f) = \sum_{j=1}^{\infty} \lambda_j^s \langle f, e_j \rangle_{L^2} e_j, \quad T^s(f) = \sum_{j=1}^{\infty} \lambda_j^s \left\langle f, \lambda_j^{\frac{1}{2}} e_j \right\rangle_{\mathcal{H}} \lambda_j^{\frac{1}{2}} e_j. \tag{4}$$

The interpolation space $[\mathcal{H}]^s$ is defined as

$$[\mathcal{H}]^s := \left\{ \sum_{j=1}^{\infty} a_j \lambda_j^{s/2} e_j : \sum_j^{\infty} a_j^2 < \infty \right\} \subseteq L^2(\mathcal{X}) \tag{5}$$

It is easy to show that $[\mathcal{H}]^s$ is also a separable Hilbert space with orthogonal basis $\{\lambda_i^{s/2} e_i\}_{i \in N}$. Specially, we have $[\mathcal{H}]^0 \subseteq L^2(\mathcal{X}, \mu)$, $[\mathcal{H}]^1 = \mathcal{H}$ and $[\mathcal{H}]^{s_2} \subsetneq [\mathcal{H}]^{s_1} \subsetneq [\mathcal{H}]^0$ for any numbers $0 < s_1 < s_2$. For the functions in $[\mathcal{H}]^s$ with larger $s$, we say they have higher (relative) interpolation smoothness with respect to the RKHS (the kernel).

## 2.2 Fractional Sobolev Space and Sobolev RKHS

For $m \in \mathbb{N}$, we denote the usual Sobolev space $W^{m,2}(\mathcal{X})$ by $H^m(\mathcal{X})$ and $L^2(\mathcal{X})$ by $H^0(\mathcal{X})$. Then the (fractional) Sobolev space for any real number $r > 0$ can be defined through the real interpolation

$$H^r(\mathcal{X}) := \left( L^2(\mathcal{X}), H^m(\mathcal{X}) \right)_{\frac{r}{m}, 2}$$

where $m := min\{k \in \mathbb{N} : k > r\}$.

It is well known that when $r > d/2$, $H^r$ is a separable RKHS with respect to a bounded kernel and the corresponding eigenvalue decay rate (EDR) is $\beta = 2r/d$. Furthermore, the interpolation space of $H^r(\mathcal{X})$ under Lebesgue measure is given by

$$[H^r(\mathcal{X})]^s = H^{rs}(\mathcal{X}). \tag{6}$$

It follows that given a Sobolev RKHS $\mathcal{H} = H^r$ for $r > d/2$, if $f \in H^a$ for any $a > 0$, one can find that $f \in [\mathcal{H}]^s$ with $s = a/r$. Thus, in this paper, we will assume that the Bayes classifier $f_\rho^*$ is in the interpolation of the Sobolev RKHS $[\mathcal{H}]^s$.

## 2.3 Kernel Classifiers: Spectra Algorithm

We then introduce a more general framework known as spectra algorithm (Rosasco et al., 2005; Caponnetto, 2006; Bauer et al., 2007). We define the filter function and the spectral algorithms as follows:

**Definition 1** (Filter function). *Let $\left\{ \varphi_\nu : \left[ 0, \kappa^2 \right] \to R^+ \mid \nu \in \Gamma \subseteq R^+ \right\}$ be a class of functions and $\psi_\nu(z) = 1 - z\varphi_\nu(z)$. If $\varphi_\nu$ and $\psi_\nu$ satisfy:*

- *$\forall \alpha \in [0, 1]$, we have $\quad \sup_{z \in [0, \kappa^2]} z^\alpha \varphi_\nu(z) \leq E\nu^{1-\alpha}, \quad \forall \nu \in \Gamma$;*

- *$\exists \tau \geq 1$ s.t. $\forall \alpha \in [0, \tau]$, we have $\quad \sup_{z \in [0, \kappa^2]} |\psi_\nu(z)| z^\alpha \leq F_\tau \nu^{-\alpha}, \quad \forall \nu \in \Gamma$,*

*where $E, F_\tau$ are absolute constants, then we call $\varphi_\nu$ a filter function. We refer to $\nu$ as the regularization parameter and $\tau$ as the qualification.*

**Definition 2** (spectral algorithm). *Let $\varphi_\nu$ be a filter function index with $\nu > 0$. Given the samples $Z$, a spectral algorithm produces an estimator of $f_\rho^*$ given by $\hat{f}_\nu = \varphi_\nu(T_X) g_Z$.*

The following example shows that $\hat{f}_t(x)$ can be formulated by the spectral algorithms.

**Example 1** (Classifier with Gradient flow). *The filter function of gradient flow $\varphi_\nu$ can be defined as $\varphi_\nu^{gf}(z) = \frac{1-e^{-\nu z}}{z}$. The qualification $\tau$ could be any positive number, $E = 1$ and $F_\tau = (\tau/e)^\tau$. So that for a test input $x$, the predicted output is given by $\hat{y} = \text{sign}(\hat{f}_\nu(x))$.*

Other spectral algorithms consist of kernel ridge regression, spectral cut-off, iterated Tikhonov, and so on. For more examples, we refer to Gerfo et al. (2008). Spectral algorithms differ in $\varphi_\nu(z)$ and $\psi_\nu(z)$, which is corresponding to saturation effect defined in Gerfo et al. (2008). Moreover, Li et al. (2023b) gives a thorough analysis of the saturation effect for kernel ridge regression.

**Notations.** Denote $B(x,r)$ as a ball, and $\lambda[B(x,r)]$ is denoted as the Lebesgue measure of $B(x,r)$. We use $\|\cdot\|_{\mathscr{B}(B_1,B_2)}$ to denote the operator norm of a bounded linear operator from a Banach space $B_1$ to $B_2$, i.e., $\|A\|_{\mathscr{B}(B_1,B_2)} = \sup_{\|f\|_{B_1}=1} \|Af\|_{B_2}$. Without bringing ambiguity, we will briefly denote the operator norm as $\|\cdot\|$. In addition, we use $\text{tr } A$ and $\|A\|_1$ to denote the trace and the trace norm of an operator. We use $\|A\|_2$ to denote the Hilbert-Schmidt norm.

# 3 MAIN RESULTS

## 3.1 ASSUMPTIONS

This subsection lists the standard assumptions for general RKHS $\mathcal{H}$ and clarifies how these assumptions correspond to properties of Sobolev RKHS.

**Assumption 1** (Source condition). *For $s > 0$, there is a constant $B > 0$ such that $f_\rho^* \in [\mathcal{H}]^s$ and $\|f_\rho^*\|_{[\mathcal{H}]^s} \leq B$.*

This assumption is weak since $s$ can be small. However, functions in $[\mathcal{H}]^s$ with smaller $s$ are less smooth, which will be harder for an algorithm to estimate.

**Assumption 2** (Eigenvalue Decay Rate (EDR)). *The EDR of the eigenvalues $\{\lambda_j\}$ associated to the kernel $K$ is $\beta > 1$, i.e.,*

$$cj^{-\beta} \leq \lambda_j \leq Cj^{-\beta} \tag{7}$$

*for some positive constants $c$ and $C$.*

Note that the eigenvalues $\lambda_i$ and EDR are only determined by the marginal distribution $\mu$ and the RKHS $\mathcal{H}$. For Sobolev RKHS $H^r$ equipped with Lebesgue measure $\nu$ and bounded domain with smooth boundary $\mathcal{X} \subseteq R^d$, it is well known that when $r > d/2$, $H^r$ is a separable RKHS with respect to a bounded kernel and the corresponding eigenvalue decay rate (EDR) is $\beta = 2r/d$ (Edmunds & Triebel, 1996).

Our next assumption is the embedding index. First, we give the definition of embedding property Fischer & Steinwart (2020): For $0 < \alpha < 1$, there is a constant $A > 0$ with $\|[H]_\nu^\alpha \hookrightarrow L_\infty(\nu)\| \leq A$. This means $[\mathcal{H}]^\alpha$ is continuously embedded into $L_\infty(\nu)$ and the operator norm of the embedding is bounded by $A$. The larger $\alpha$ is, the weaker the embedding property is.

**Assumption 3** (Embedding index). *Suppose that there exists $\alpha_0 > 0$, such that*

$$\alpha_0 = \inf\left\{\alpha \in [\frac{1}{\beta}, 1] : \|[\mathcal{H}]^\alpha \hookrightarrow L^\infty(\mathcal{X}, \mu)\| < \infty\right\},$$

*and we refer to $\alpha_0$ as the embedding index of an RKHS $\mathcal{H}$.*

This assumption directly implies that all the functions in $[\mathcal{H}]^\alpha$ are $\mu$-a.e bounded for $\alpha > \alpha_0$. Moreover, we will clarify this assumption for Sobolev kernels and dot-product kernels on $\mathbb{S}^{d-1}$ in the appendix.

## 3.2 MINIMAX OPTIMALITY OF KERNEL CLASSIFIERS

This subsection presents our main results on the minimax optimality of kernel classifiers. We first establish a minimax lower bound for the Sobolev RKHS $H^r(\mathcal{X})$ under the source condition (Assumption 1). We then provide an upper bound based on Assumptions 1, 2, and 3, and we clarify that

the Sobolev RKHS satisfies these assumptions. As a result, we demonstrate that the Sobolev kernel classifier is minimax rate optimal.

**Theorem 1** (Lower Bound). *Suppose $f_\rho^* \in [H^r(\mathcal{X})]^s$ for $s > 0$, where $H^r$ is the Sobolev RKHS. For all learning methods $\hat{f}$, for any fixed $\delta \in (0, 1)$, when $n$ is sufficiently large, there is a distribution $\rho \in \mathcal{P}$ such that, with probability at least $1 - \delta$, we have*

$$\mathcal{E}(\hat{f}) \geq C\delta n^{-\frac{s\beta}{2(s\beta+1)}}, \tag{8}$$

*where $C$ is a universal constant.*

Theorem 1 shows the minimax lower bound on the classification excess risk over the interpolation space of the Sobolev RKHS. Theorem 1 also establishes a minimax lower bound at the rate of $n^{-\frac{a}{2a+d}}$ for the Sobolev space $H^a$ with $a = rs$. Yang (1999) illustrated a similar result of the minimax lower bound for Besov spaces. However, the result has only been proved for $d = 1$ by Kerkyacharian & Picard (1992) and the case for $d > 1$ remains unresolved.

The following theorem presents an upper bound for the kernel classifier.

**Theorem 2** (Upper Bound). *Suppose that Assumptions 1, 2, and 3 hold for $0 < s \leq 2\tau$, where $\tau$ is the qualification of the filter function. By choosing $\nu \asymp n^{\frac{\beta}{s\beta+1}}$, for any fixed $\delta \in (0, 1)$, when $n$ is sufficiently large, with probability at least $1 - \delta$, we have*

$$\mathcal{E}(\hat{f}_\nu) \leq C \left( \ln \frac{4}{\delta} \right) n^{-\frac{s\beta}{2(s\beta+1)}} \tag{9}$$

*where $C$ is a constant independent of $n$ and $\delta$.*

Combined with Theorem 1, Theorem 2 shows that by choosing a proper early-stopping time, the Sobolev kernel classifier is minimax rate optimal. Moreover, given the kernel and the decay rate $\beta$, the optimal rate is mainly affected by the smoothness $s$ of $f_\rho^*$ with respect to the kernel. Thus, in Section 5, we will introduce how to estimate the smoothness of functions or datasets given a specific kernel.

We emphasize that Theorem 2 can be applied to any general RKHS with an embedding index $\alpha_0 = 1/\beta$, such as an RKHS with a shift-invariant periodic kernel and an RKHS with a dot-product kernel. Thanks to the uniform convergence of overparameterized neural networks (Lai et al., 2023; Li et al., 2023a), Theorem 2 can also be applied to analyze the generalization error of the neural network classifiers. We will discuss this application in the next section.

## 4 APPLICATIONS IN NEURAL NETWORKS

Suppose that we have observed $n$ i.i.d. samples $\{X_i, Y_i\}_{i=1}^n$ from $\rho$. For simplicity, we further assume that the marginal distribution $\mu$ of $\rho$ is the uniform distribution on the unit sphere $\mathbb{S}^{d-1}$. We use a neural network with $L$ hidden layers and width $m$ to perform the classification on $\{X_i, Y_i\}_{i=1}^n$. The network model $f(x; \theta)$ and the resulting prediction are given by the following equations

$$h^0(x) = x, \quad h^l(x) = \sqrt{\frac{2}{m}} \sigma(W^{l-1} h^{l-1}(x)), \quad l = 1, ..., L$$
$$f(x; \theta) = W^L h^L(x) \quad \text{and} \quad \hat{y} = \text{sign}(f(x; \theta)), \tag{10}$$

where $h^l$ represents the hidden layer, $\sigma(x) := \max(x, 0)$ is the ReLU activation (applied element-wise), $W^0 \in \mathbb{R}^{m \times d}$ and $W^l \in \mathbb{R}^{m \times m}$ are the parameters of the model. We use $\theta$ to represent the collection of all parameters flatten as a column vector. With the mirrored initialization (shown in Li et al. (2023a)), we consider the training process given by the gradient flow $\dot{\theta} = -\partial L(\theta)/\partial \theta$, where the squared loss function is adopted $L(\theta) = \frac{1}{2n} \sum_{i=1}^n (Y_i - f(X_i, \theta))^2$.

The consideration for this choice of loss function is that the squared loss function is robust for optimization and more suitable for hard learning scenarios (Hui & Belkin (2020); Demirkaya et al. (2020); Kornblith et al. (2020)). Hui & Belkin (2020) showed that the square loss function has been shown to perform well in modern classification tasks, especially in natural language processing while Kornblith et al. (2020) presented the out-of-distribution robustness of the square loss function.

When the network is overparameterized, Li et al. (2023a) showed that the trained network $f(x; \theta)$ can be approximated by a kernel gradient method with respect to the following neural tangent kernel

$$K_{ntk}(x, x') = \sum_{r=0}^{L} \kappa_1^{(r)}(\bar{u}) \prod_{s=r}^{L-1} \kappa_0(\kappa_1^{(s)}(\bar{u})) \tag{11}$$

where $\bar{u} = \langle x, x' \rangle$, $\kappa_1^{(p)} = \kappa_1 \underbrace{\circ \cdots \circ}_{p \text{ times}} \kappa_1$ represents $p$ times composition of $\kappa_1$ and $\kappa_1^{(0)}(u) = u$ by convention; if $r = L$, the product $\prod_{s=r}^{L-1}$ is understood to be 1. Denote $Y_{[n]} = (Y_1, ..., Y_n)^T$, $K(X_{[n]}, X_{[n]})$ as an $n \times n$ matrix of $(K(X_i, X_j))_{i,j \in [n]}$ and $\lambda_{min} = \lambda_{min}(K(X_{[n]}, X_{[n]}))$ The following proposition shows the uniform convergence of $f(x; \theta)$.

**Proposition 1** (Theorem 1 in Li et al. (2023a)). *Suppose $x \in \mathbb{S}^{d-1}$. For any $\epsilon > 0$, any hidden layer $L \geq 2$, and $\delta \in (0, 1)$, when the width $m \geq \text{poly}\left(n, \lambda_{min}^{-1}, ||Y_{[n]}||_2, \ln(1/\delta), \ln(1/\epsilon)\right)$, with probability at least $1 - \delta$ with respect to random initialization, we have*

$$\sup_{t \geq 0} \sup_{x \in \mathcal{X}} |f_t(x; \theta) - f_t^{ntk}(x)| \leq \epsilon.$$

*where $f_t^{ntk}(x)$ is defined as in Example 1 but with the kernel $K_{ntk}$.*

Theorem G.5 in Haas et al. (2023) showed that the RKHS of the NTK on $\mathbb{S}^{d-1}$ is a Sobolev space. Moreover, the kernel $K_{ntk}$ is a dot-product kernel satisfying a polynomial eigenvalue decay $\beta = d/(d-1)$. Thus, we can obtain the following corollary by combining Theorem 2 and Proposition 1.

**Corollary 1.** *Suppose that $x \in \mathbb{S}^{d-1}$ and Assumption 1 holds for $\mathcal{H}$ being the RKHS of the kernel $K_{ntk}$ and $s > 0$. Suppose $t \asymp n^{\frac{\beta}{s\beta+1}}$. For any fixed $\delta \in (0, 1)$, when $m \geq \text{poly}\left(n, \lambda_{min}^{-1}, ||Y_{[n]}||_2, \ln(1/\delta)\right)$ and $n$ is sufficiently large, with probability at least $1 - \delta$, we have*

$$\mathcal{E}(f_t(x; \theta)) \leq C \left(\ln \frac{4}{\delta}\right) n^{-\frac{s\beta}{2(s\beta+1)}} \tag{12}$$

*where $C$ is a constant independent of $n$ and $\delta$.*

This corollary shows that the generalization error of a fine-tuned, overparameterized neural network classifier converges at the rate of $n^{-\frac{s\beta}{2(s\beta+1)}}$. This result also highlights the need for additional efforts to understand the smoothness of real datasets with respect to the neural tangent kernel. A larger value of $s$ corresponds to a faster convergence rate, indicating the possibility of better generalization performance. Determination of the smoothness parameter $s$ will allow us to assess the performance of an overparameterized neural network classifier on a specific dataset.

## 5 ESTIMATION OF SMOOTHNESS

In this section, we provide a simple example to illustrate how to determine the relative smoothness $s$ of the ground-truth function with respect to the kernel. Then we introduce a simple method to estimate $s$ with noise and apply the method to real datasets with respect to the NTK.

**Determination of $s$.** Suppose that $\mathcal{X} \in [0, 1]$ and the marginal distribution $\mu_{\mathcal{X}}$ is a uniform distribution on $[0, 1]$. We consider the min kernel $K_{min}(x, x') = \min(x, x')$ (Wainwright, 2019) and denote by $\mathcal{H}_{min}$ the corresponding RKHS. The eigenvalues and the eigenfunctions of $\mathcal{H}_{min}$ are

$$\lambda_j = \left(\frac{2j-1}{2}\pi\right)^{-2}, \quad e_j(x) = \sqrt{2}\sin(\frac{2j-1}{2}\pi x), \quad j \geq 1. \tag{13}$$

Thus, the EDR is $\beta = 2$. For illustration, we consider the ground true function $f^*(x) = \cos(2\pi x)$. Suppose $f^*(x) = \sum_j^\infty f_j e_j(x)$, then we have $f_j = \sqrt{2} \int_0^1 \cos(2\pi x) \sin(\frac{2j-1}{2}\pi x) dx \asymp j^{-1}$. Thus, $f_j \asymp j^{-r}$ where $r = 1$. By the definition of the interpolation space, we have $s = \frac{2r-1}{\beta} = 0.5$.

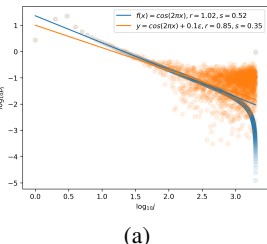 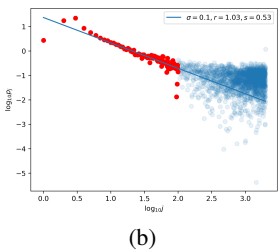 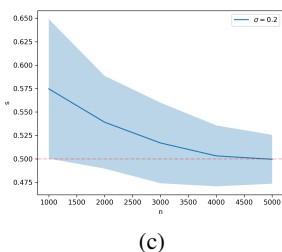

(a)                    (b)                    (c)

Figure 1: Experiments for estimating the smoothness parameter $s$ in regression settings. (a) Naive estimation based on $2,000$ sample points for $\sigma = 0$ (blue) and $\sigma = 0.1$ (orange). (b) Truncation Estimation based on $2,000$ sample points with truncation point $100$. In both plots (a) and (b), the $x$-axis is the logarithmic index $j$ and the $y$-axis is the logarithmic $p_j$.(c) Truncation Estimation across various values of sample size $n$, each repeated 50 times. The blue line represents the average of estimates, the shaded area represents one standard deviation, and the true value is indicated by the orange dashed line.

**Estimation of $s$ in regression.**    To better understand the estimation process, we first consider regression settings where the noises have an explicit form and we then consider classification settings. Suppose that we have $n$ i.i.d. samples of $X_{[n]} = [X_1, ..., X_n]^\top$ and $Y_{[n]} = [Y_1, ..., Y_n]^\top \in$ from $Y_i = f^*(X_i) + \sigma \epsilon_i$, where $\epsilon_i \sim \mathcal{N}(0, 1)$.

We start with a naive estimation method. Let $K_{min}(X_{[n]}, X_{[n]})$ be the kernel matrix. Suppose the eigendecomposition is given by $K_{min}(X_{[n]}, X_{[n]}) = V\Sigma V^\top$, where $V = [v_1, ..., v_n]$ is the eigenvector matrix, $v_i$'s are the eigenvectors, and $\Sigma$ is the diagonal matrix of the eigenvalues. We can estimate $r$ by estimating the decay rate of $p_j$, where $p_j = Y_{[n]}^\top v_j$. To visualize the convergence rate $r$, we perform logarithmic least-squares to fit $p_j$ with respect to the index $j$ and display the values of the slope $r$ and the smoothness parameter $s$.

For $\sigma = 0$, $r$ can be accurately estimated by the above naive method since there is no noise in $Y_i$'s. The blue line and dots in Figure 1 (a) present the estimation of $s$ in this case, where the estimate is around the true value $0.5$. However, for $\sigma = 0.1$, the naive estimation is not accurate, as shown by the orange line and dots in Figure 1 (a).

To improve the accuracy of the estimation, we introduce a simple modification called *Truncation Estimation*, described as follows. We select some fixed integer as a truncation point and estimate the decay rate of $p_j$ up to the truncation point. For the example with $\sigma = 0.1$, we choose the truncation point $100$ and the result is shown in Figure 1 (b). We observe that the estimation becomes much more accurate than the naive estimation, with an estimate of $s = 0.53$ not too far away from the true value $0.5$. In general, noise in the data can worsen the estimation accuracy, while increasing the sample size can improve the accuracy and robustness of the estimation. In Figure 1 (c), we show the result for estimating $s$ in repeated experiments with more noisy data ($\sigma = 0.2$), where we observe that as the sample size $n$ increases, the estimation becomes accurate.

**Estimation of $s$ in classification.**    Now we consider the classification settings, where the population is given by $\mathbb{P}(Y = 1 | X = x) = (f^*(x) + 1)/2$. Unlike regression problems, the variance of the noise $\epsilon = y - f^*(x)$ is determined by $f^*(x)$ and may not be negligible. Nonetheless, in classification problems, we can still estimate the smoothness parameter $s$ using Truncation Estimation, thanks to the fact that increasing the sample size can improve its performance. The results are shown in Figure 2, where we can indeed make similar observations to those in Figure 1 (b) and (c).

As an application of Truncation Estimation, we estimate the relative smoothness of real data sets with respect to the NTK defined in equation 11. The results are shown in Table 1. We can see that with respect to the NTK, MNIST has the largest relative smoothness while CIFAR-10 has the smallest one. This result aligns with the common knowledge that MNIST is the easiest dataset while CIFAR-10 is the most difficult one of these three datasets.

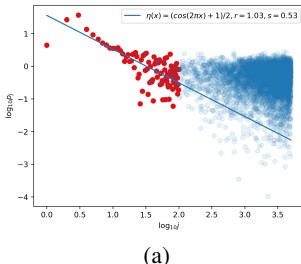 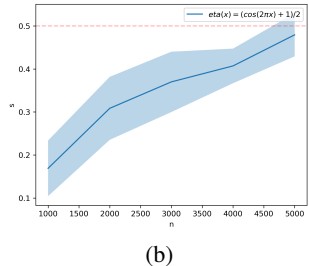

(a)                                        (b)

Figure 2: Experiments for estimating the smoothness parameter $s$ in classification settings. (a) The experiment uses $5,000$ sample points and the truncation point is $100$. (b) Truncation Estimation across various values of sample size $n$, each repeated 50 times. The blue line represents the average of estimates, the shaded area represents one standard deviation, and the true value is indicated by the orange dashed line.

| Kernel | MNIST | Fashion-MNIST | CIFAR-10 |
|--------|-------|---------------|----------|
| NTK-1 | 0.4862 (0.0824) | 0.4417 (0.0934) | 0.1992 (0.0724) |
| NTK-2 | 0.4871 (0.0793) | 0.4326 (0.0875) | 0.2047 (0.0831) |
| NTK-3 | 0.4865 (0.0815) | 0.4372 (0.0768) | 0.1965 (0.0795) |

Table 1: Truncation Estimation of the relative smoothness $s$ of different real data sets with different NTKs. $NTK - L$ indicates the $L$-hidden-layer NTK. We only consider two classes of labels for each dataset: Label 1 and 7 for MNIST, trousers and sneakers for Fashion-MNIST, cars and horses for CIFAR-10. We randomly select 5,000 data points and choose the truncation point $100$ to estimate $s$. For each dataset and each kernel, we repeat 50 times and the standard deviation is in parentheses.

**Limitations** The misspecified spectral algorithms (assuming $f_\rho^* \in [\mathcal{H}]^s$) are studied since 2009 (e.g., Steinwart et al. (2009); Dicker et al. (2017); Pillaud-Vivien et al. (2018); Fischer & Steinwart (2020); Zhang et al. (2023)). However, to the best of our knowledge, there is barely any result on the estimation of the smoothness $s$. This paper is the first to propose the $s$ estimation method even though the method is more susceptible to noise when the sample size is not enough or $f^*$ has more complex structures. For example, if $f^* = \sum_{j=1}^\infty f_j e_j(x)$, where $f_j^2 = j^{-s_1\beta-1}$ when $j$ is odd and $f_j^2 = j^{-s_2\beta-1}$ when $j$ is even ($s_1 > s_2$). For the kernel $K$ with EDR $\beta$, $f_\rho^* \in [\mathcal{H}]^{s_2}$ instead of $[\mathcal{H}]^{s_1}$ or $[\mathcal{H}]^s$ for some $s \in (s_2, s_1)$. In this mixed smoothness case, our method tends to give an estimation $\hat{s} \in (s_2, s_1)$. A more detailed discussion of the limitations is presented in the appendix. We will try to find more accurate $s$ estimation methods for general situations in the near future.

## 6 DISCUSSION

In this paper, we study the generalization error of kernel classifiers in Sobolev space (the interpolation of the Sobolev RKHS). We show the optimality of kernel classifiers under the assumption that the ground true function is in the interpolation of RKHS with the kernel. The minimax optimal rate is $n^{-s\beta/2(s\beta+1)}$, where $s$ is the smoothness parameter of the ground true function. Building upon the connection between kernel methods and neural networks, we obtain an upper bound on the generalization error of overparameterized neural network classifiers. To make our theoretical result more applicable to real problems, we propose a simple method called Truncation Estimation to estimate the relative smoothness $s$. Using this method, we examine the relative smoothness of three real datasets, including MNIST, Fashion-MNIST and CIFAR-10. Our results confirm that among these three datasets, MNIST is the simplest for classification using NTK classifiers while CIFAR-10 is the hardest.

ACKNOWLEDGMENTS

Lin's research was supported in part by the National Natural Science Foundation of China (Grant 92370122, Grant 11971257). This work has been supported by the New Cornerstone Science Foundation.

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
