(A.1 and A.2) and then present the minimax lower bound (A.3). Before the proof, We list again the standard assumptions for general RKHS $\mathcal{H}$ in this section.

**Assumption 4** (Source condition). *For $s > 0$, there is a constant $B > 0$ such that $f_\rho^* \in [\mathcal{H}]^s$ and*

$$\|f_\rho^*\|_{[\mathcal{H}]^s} \le B.$$

**Assumption 5** (Eigenvalue Decay Rate (EDR)). *The EDR of the eigenvalues $\{\lambda_j\}$ associated to the kernel $K$ is $\beta$, i.e.,*

$$cj^{-\beta} \le \lambda_j \le Cj^{-\beta} \tag{14}$$

*for some positive constants $c$ and $C$ and $\beta > 1$.*

**Assumption 6** (Embedding index). *Suppose that there exists $\alpha_0 > 0$, such that*

$$\alpha_0 = \inf \left\{ \alpha \in [\frac{1}{\beta}, 1] : \|[\mathcal{H}]^\alpha \hookrightarrow L^\infty(\mathcal{X}, \mu)\| < \infty \right\},$$

*and we refer to $\alpha_0$ as the embedding index of an RKHS $\mathcal{H}$.*

Define the sampling operator $K_x : \mathbb{R} \to \mathcal{H}$, $y \mapsto yK(x, \cdot)$ and its adjoint operator $K_x^* : \mathcal{H} \to \mathbb{R}$, $f \mapsto f(x)$. Further, we define the sample covariance operator $T_X : \mathcal{H} \to \mathcal{H}$ as

$$T_X := \frac{1}{n} \sum_{i=1}^n K_{X_i} K_{X_i}^*.$$

Then we know that $\|T_X\| \le \|T_X\|_1 \le \kappa^2$, where $\|\cdot\|$ denotes the operator norm and $\|\cdot\|_1$ denotes the trace norm. Further, define the sample basis function

$$g_Z := \frac{1}{n} \sum_{i=1}^n K_{X_i} Y_i \in \mathcal{H}.$$

We also introduce a more general framework known as spectra algorithm Rosasco et al. (2005); Caponnetto (2006); Bauer et al. (2007). We define the filter function and the spectral algorithms as follows:

**Definition 3** (Filter function). *Let $\left\{\varphi_\nu : \left[0, \kappa^2\right] \to R^+ \mid \nu \in \Gamma \subseteq R^+\right\}$ be a class of functions and $\psi_\nu(z) = 1 - z\varphi_\nu(z)$. If $\varphi_\nu$ and $\psi_\nu$ satisfy:*

- *$\forall \alpha \in [0, 1]$, we have*

$$\sup_{z \in [0, \kappa^2]} z^\alpha \varphi_\nu(z) \leq E\nu^{1-\alpha}, \quad \forall \nu \in \Gamma; \tag{15}$$

- *$\exists \tau \geq 1$ s.t. $\forall \alpha \in [0, \tau]$, we have*

$$\sup_{z \in [0, \kappa^2]} |\psi_\nu(z)| z^\alpha \leq F_\tau \nu^{-\alpha}, \quad \forall \nu \in \Gamma, \tag{16}$$

*where $E, F_\tau$ are absolute constants, then we call $\varphi_\nu$ a filter function. We refer to $\nu$ as the regularization parameter and $\tau$ as the qualification.*

**Definition 4** (spectral algorithm). *Let $\varphi_\nu$ be a filter function index with $\nu > 0$. Given the samples $Z$, the spectral algorithm produces an estimator of $f_\rho^*$ given by*

$$\hat{f}_\nu = \varphi_\nu(T_X) g_Z. \tag{17}$$

## A.1 SOME BOUNDS

Throughout the proof, we denote

$$T_\nu = T + \nu^{-1}; \quad T_{X\nu} = T_X + \nu^{-1}$$

where $\nu$ is the regularization parameter. In addition, we denote $L^2(X, \mu)$ as $L^2$, $L^\infty(X, \mu)$ as $L^\infty$ for brevity throughout the proof. We use $a_n \asymp b_n$ to denote that there exist constants $c$ and $C$ such that $ca_n \leq b_n \leq Ca_n, \forall n = 1, 2, \cdots$; use $a_n \lesssim b_n$ to denote that there exists an constant $C$ such that $a_n \leq Cb_n, \forall n = 1, 2, \cdots$ In addition, denote the effective dimension as

$$N(\nu) = \text{tr}\left(T\left(T + \nu^{-1}\right)^{-1}\right) = \sum_{i \in N} \frac{\lambda_i}{\lambda_i + \nu^{-1}}$$

**Lemma 1.** *Suppose $\nu > 1$. If $\lambda_i \asymp i^{-\beta}$, we have*

$$N(\nu) \asymp \nu^{\frac{1}{\beta}}.$$

*Proof.* Since $ci^{-\beta} \leq \lambda_i \leq Ci^{-\beta}$, we have

$$N(\nu) = \sum_{i=1}^\infty \frac{\lambda_i}{\lambda_i + \nu^{-1}} \leq \sum_{i=1}^\infty \frac{Ci^{-\beta}}{Ci^{-\beta} + \nu^{-1}} = \sum_{i=1}^\infty \frac{C}{C + \nu^{-1}i^\beta}$$

$$\leq \int_0^\infty \frac{C}{\nu^{-1}x^\beta + C} dx = \nu^{\frac{1}{\beta}} \int_0^\infty \frac{C}{y^\beta + C} dy \leq C_1 \nu^{\frac{1}{\beta}}$$

for some constant $C_1$. Since $\nu > 1$, the proof for the lower bound can be obtained similarly.

$\square$

### A.1.1 APPROXIMATION ERROR

Recall that we have defined the sample basis function $g_Z$ and the spectral algorithm $\hat{f}_\nu$. We also need the following notations: define the expectation of $g_Z$ as

$$g = Eg_Z = \int_{\mathcal{X}} K(x, \cdot) f_\rho^*(x) d\mu(x) = S_k^* f_\rho^* \in \mathcal{H},$$

and

$$f_\nu = \varphi_\nu(T)g = \varphi_\nu(T)S_k^* f_\rho^*$$

The following conclusion based on Zhang et al. (2023) bounds the $L^2$-norm of $f_\nu - f_\rho^*$ for spectral algorithm:

**Lemma 2.** *Suppose that Assumption 4 holds for $0 < s \leq 2\tau$. Then for any $\nu > 0$, we have*

$$\left\| f_\nu - f_\rho^* \right\|_{L^2} \leq F_\tau R \nu^{-\frac{s}{2}}.$$

*Proof.* Because $f_\rho^* \in [\mathcal{H}]^s$, we assume $f_\rho^* = L^{\frac{s}{2}} g_0$ for some $g_0 \in L^2(P)$, so that $\|g_0\|_{L^2} \leq B$ by Assumption 4. By the definition of $f_\nu$, we have:

$$
\begin{aligned}
\|f_\nu - f_\rho^*\|_{L^2} &= \|\varphi_\nu(T) S_k^* f_\rho^* - f_\rho^*\|_{L^2} \\
&= \|(\varphi_\nu(L)L - Id)L^{\frac{s}{2}} g_0\|_{L^2} \\
&\leq \|(\psi_\nu(L)L^{\frac{s}{2}} g_0\|_{L^2} \\
&\leq F_r B \nu^{-\frac{s}{2}}.
\end{aligned}
$$

Where the second equality holds by the definition of natural embedding inclusion operator $S_k$, and $Id$ denotes identity mapping. The first inequality holds because of the definition of $\psi_\nu$ and the second inequality holds for equation 16. $\square$

### A.1.2 Estimation error

We rewrite the estimation error as follows

$$
\begin{aligned}
\left\| \hat{f}_\nu - f_\nu \right\|_{L^2} &= \left\| T^{\frac{1}{2}} \left( \hat{f}_\nu - f_\nu \right) \right\|_{\mathcal{H}} \\
&= \left\| T^{\frac{1}{2}} T_\nu^{-\frac{1}{2}} \cdot T_\nu^{\frac{1}{2}} T_{X\nu}^{-\frac{1}{2}} \cdot T_{X\nu}^{\frac{1}{2}} \left( \hat{f}_\nu - f_\nu \right) \right\|_{\mathcal{H}} \qquad (18) \\
&\leq \left\| T^{\frac{1}{2}} T_\nu^{-\frac{1}{2}} \right\|_{\mathscr{B}(\mathcal{H})} \cdot \left\| T_\nu^{\frac{1}{2}} T_{X\nu}^{-\frac{1}{2}} \right\|_{\mathscr{B}(\mathcal{H})} \cdot \left\| T_{X\nu}^{\frac{1}{2}} \left( \hat{f}_\nu - f_\nu \right) \right\|_{\mathcal{H}}.
\end{aligned}
$$

**Step 1.** The first part can be bounded by the following lemma, whose proof is simple and omitted.
**Lemma 3.**

$$\left\| T^{\frac{1}{2}} \left( T_\nu \right)^{-1/2} \right\|^2 = \sup_{i \geq 1} \frac{\lambda_i}{\lambda_i + \nu^{-1}} \leq 1.$$

For the second part, we recall a result (Fischer & Steinwart, 2020, Lemma 11).

**Lemma 4** (Fischer & Steinwart (2020)). *Suppose that the RKHS $\mathcal{H}$ has the embedding index $\alpha_0$. Then for any $\alpha_0 < \alpha \leq 1$ and all $\delta \in (0, 1)$, with probability at least $1 - \delta$, we have*

$$\left\| T_\nu^{-\frac{1}{2}} \left( T - T_X \right) T_\nu^{-\frac{1}{2}} \right\| \leq \frac{4M_\alpha^2 \nu^\alpha}{3n} B + \sqrt{\frac{2M_\alpha^2 \nu^\alpha}{n}} B,$$

*where*

$$B = \ln \frac{4N(\nu) \left( \|T\| + \nu^{-1} \right)}{\delta \|T\|}.$$

**Lemma 5.** *If the sample size $n \geq 8M_\alpha^2 \nu^\alpha B$, we have:*

$$\left\| T_\nu^{\frac{1}{2}} T_{X\nu}^{-\frac{1}{2}} \right\|_{\mathscr{B}(\mathcal{H})} \leq 3$$

*holds with probability at least $1 - \delta$.*

*Proof.* We calculate $T_\nu^{\frac{1}{2}} T_{X\nu}^{-\frac{1}{2}}$ directly, and combined with Lemma 4

$$
\begin{aligned}
T_{X\nu} &= T_X + \nu^{-1} = T_X - T + T + \nu^{-1} \\
&= (T_\nu)^{\frac{1}{2}} \left[ Id - (T_\nu)^{-\frac{1}{2}} (T - T_X)(T_\nu)^{-\frac{1}{2}} \right] (T_\nu)^{\frac{1}{2}},
\end{aligned}
$$

By Lemma 4 when $n \geq 8M_\alpha^2 \nu^\alpha B$, we have:

$$\left\| T_\nu^{-\frac{1}{2}} \left( T - T_X \right) T_\nu^{-\frac{1}{2}} \right\| \leq \frac{4}{3} \cdot \frac{1}{8} + \sqrt{\frac{1}{4}} \leq \frac{2}{3}$$

holds with probability at least $1 - \delta$. So the Neumann series gives us the following bound

$$
\begin{aligned}
\left\| T_\nu^{\frac{1}{2}} T_{X\nu}^{-\frac{1}{2}} \right\|_{\mathscr{B}(\mathcal{H})} &= \left\| \left( Id - (T_\nu)^{-\frac{1}{2}} (T - T_X)(T_\nu)^{-\frac{1}{2}} \right)^{-1} \right\|_{\mathscr{B}(\mathcal{H})} \\
&\leq \sum_{k=0}^{\infty} \left\| T_\nu^{-\frac{1}{2}} (T - T_X) T_\nu^{-\frac{1}{2}} \right\|^k \\
&\leq \sum_{k=0}^{\infty} \left( \frac{2}{3} \right)^k = 3.
\end{aligned}
$$

$\square$

**Step 2.** For the third part in the last line of equation 18, we have

$$
\left\| T_{X\nu}^{\frac{1}{2}} \left( \hat{f}_\nu - f_\nu \right) \right\|_{\mathcal{H}} \leq \left\| T_{X\nu}^{\frac{1}{2}} \varphi_\nu (T_X) (g_Z - T_X f_\nu) \right\|_{\mathcal{H}} + \left\| T_{X\nu}^{\frac{1}{2}} \psi_\nu (T_X) f_\nu \right\|_{\mathcal{H}}. \tag{19}
$$

The second part of RHS in equation 19 is complicated to calculate, but its proof follows the same argument as in Step 3 of Theorem 16 in Zhang et al. (2023). Therefore, we provide the following result without proof:

**Lemma 6** (Theorem 16 in Zhang et al. (2023)). *If $s < 2\tau$, we have:*

$$
\left\| T_{X\nu}^{\frac{1}{2}} \psi_\nu (T_X) f_\nu \right\|_{\mathcal{H}} \leq 6 F_\tau E R \nu^{-\frac{s}{2}} + \Delta_1 I_{s>2}, \tag{20}
$$

*where $\Delta_1$ denotes*

$$
\Delta_1 = 32 \max \left( \frac{s-1}{2}, 1 \right) E F_\tau R \kappa^{s-1} \nu^{-\frac{1}{2}} n^{-\frac{\min(s,3)-1}{4}} \ln \frac{6}{\delta}.
$$

*We need $s < 2\tau$, because we use equation 16 to upper bound $\|T_X^{\frac{s}{2}} \psi_\nu(T_X)\|$ for every $s$.*

To bound the first term of RHS in equation 19, we begin with a lemma, whose proof is postponed to Section A.1.3.

**Lemma 7.** *Suppose that Assumption 4, 5 and 6 hold for $\frac{1}{\beta} \leq \alpha_0 < 1$, and conditional probability of $Y$ is given by*

$$
\mathbb{P}(Y = 1 | X = x) = \frac{1 + f_\rho^*(x)}{2}.
$$

*Then given $\nu > 0$, and $n \geq 1$, for any $\delta > 0$ and $\alpha > \alpha_0$, the following bound is satisfied with probability not less than $1 - \delta$*

$$
\left\| (T_\nu)^{-1/2} ((g_Z - T_X f_\nu) - (g - T f_\nu)) \right\|_{\mathcal{H}}^2 \leq \frac{128 \left( \log \frac{2}{\delta} \right)^2}{n} \left( N(\nu) + \frac{4 M_\alpha^2 \nu^\alpha}{n} \right).
$$

The next lemma provides a bound on the first term of RHS in equation 19.

**Lemma 8.** *Suppose that Assumption 4, 5 and 6 hold for $\frac{1}{\beta} \leq \alpha_0 < 1$. If the sample size $n \geq 8 M_\alpha^2 \nu^\alpha B$, where $B$ is defined in Lemma 4 and $\nu > 0$, then for any $\delta > 0$ and $\alpha > \alpha_0$, the following bound is satisfied with probability at least $1 - \delta$:*

$$
\left\| T_{X\nu}^{\frac{1}{2}} \varphi_\nu (T_X) (g_Z - T_X f_\nu) \right\|_{\mathcal{H}} \leq C \left[ \left( \log \frac{2}{\delta} \right) \left( \frac{N(\nu)^{\frac{1}{2}}}{\sqrt{n}} + \frac{2 M_\alpha \nu^{\frac{\alpha}{2}}}{n} \right) + \left\| f_\rho^* - f_\nu \right\|_{L^2} \right],
$$

*where $C$ is an absolute constant.*

*Proof.*

$$
\begin{aligned}
\left\| T_{X\nu}^{\frac{1}{2}} \varphi_\nu (T_X) (g_Z - T_X f_\nu) \right\|_{\mathcal{H}} &= \left\| T_{X\nu}^{\frac{1}{2}} \varphi_\nu (T_X) T_{X\nu}^{\frac{1}{2}} \cdot T_{X\nu}^{-\frac{1}{2}} T_\nu^{\frac{1}{2}} \cdot T_\nu^{-\frac{1}{2}} (g_Z - T_X f_\nu) \right\|_{\mathcal{H}} \\
&\leq \left\| T_{X\nu}^{\frac{1}{2}} \varphi_\nu (T_X) T_{X\nu}^{\frac{1}{2}} \right\|_{\mathscr{B}(\mathcal{H})} \cdot \left\| T_{X\nu}^{-\frac{1}{2}} T_\nu^{\frac{1}{2}} \right\|_{\mathscr{B}(\mathcal{H})} \\
&\quad \cdot \left\| T_\nu^{-\frac{1}{2}} (g_Z - T_X f_\nu) \right\|_{\mathcal{H}}.
\end{aligned} \tag{21}
$$

The first term can be upper bounded as

$$
\left\| T_{X\nu}^{\frac{1}{2}} \varphi_\nu \left( T_X \right) T_{X\nu}^{\frac{1}{2}} \right\|_{\mathscr{B}(\mathcal{H})} = \| (T_X + \nu^{-1}) \varphi_\nu \left( T_X \right) \|_{\mathscr{B}(\mathcal{H})}
$$

$$
\leq \| T_X \varphi_\nu \left( T_X \right) \|_{\mathscr{B}(\mathcal{H})} + \nu^{-1} \| \varphi_\nu \left( T_X \right) \|_{\mathscr{B}(\mathcal{H})}
$$

$$
\leq 2E
$$

because we have $z\varphi_\nu(z) \leq E$ by equation 15 and $\varphi_\nu(z) \leq E\nu$ by equation 16. Using Lemma 5, the second term can be bounded as

$$
\left\| T_{X\nu}^{-\frac{1}{2}} T_\nu^{\frac{1}{2}} \right\|_{\mathscr{B}(\mathcal{H})} \leq 3.
$$

It remains to bound the third term. Lemma 7 shows that

$$
\left\| (T_\nu)^{-1/2} \left( (g_Z - T_X f_\nu) - (g - T f_\nu) \right) \right\|_{\mathcal{H}}^2 \leq \frac{128 \left( \log \frac{2}{\delta} \right)^2}{n} \left( N(\nu) + \frac{4 M_\alpha^2 \nu^\alpha}{n} \right). \tag{22}
$$

Thus, we have

$$
\left\| T_\nu^{-\frac{1}{2}} \left( g_Z - T_X f_\nu \right) \right\|_{\mathcal{H}} \leq \left\| T_\nu^{-\frac{1}{2}} \left[ (g_Z - T_X f_\nu) - (g - T f_\nu) \right] \right\|_{\mathcal{H}} + \left\| T_\nu^{-\frac{1}{2}} \left( g - T f_\nu \right) \right\|_{\mathcal{H}}
$$

$$
\leq 8\sqrt{2} \left( \log \frac{2}{\delta} \right) \left( \frac{N(\nu)^{\frac{1}{2}}}{\sqrt{n}} + \frac{2 M_\alpha \nu^{\frac{\alpha}{2}}}{n} \right) + \left\| T_\nu^{-\frac{1}{2}} S_k^* \right\|_{\mathscr{B}(L^2, \mathcal{H})} \left\| f_\rho^* - f_\nu \right\|_{L^2}
$$

$$
\leq 8\sqrt{2} \left( \log \frac{2}{\delta} \right) \left( \frac{N(\nu)^{\frac{1}{2}}}{\sqrt{n}} + \frac{2 M_\alpha \nu^{\frac{\alpha}{2}}}{n} \right) + \left\| f_\rho^* - f_\nu \right\|_{L^2},
$$

where the second term is approximation error that has been referred to in Lemma 2. $\qquad\square$

**Step 3.** Now we combine the bounds for the three parts of estimation error in equation 18: the first two parts are corresponding to Lemma 3 and 5 respectively, and the third part is corresponding to equation 19, equation 20, and Lemma 8. Then based on Assumptions the same as Lemma 8, combined with $s < 2\tau$, we conclude that for any $\delta > 0$, it holds with probability at least $1 - \delta$

$$
\left\| \hat{f}_\nu - f_\nu \right\|_{L^2} \leq C \left( \log \frac{2}{\delta} \right) \left( \frac{N(\nu)^{\frac{1}{2}}}{\sqrt{n}} + \frac{2 M_\alpha \nu^{\frac{\alpha}{2}}}{n} \right)
$$
$$
+ \left\| f_\rho^* - f_\nu \right\|_{L^2} + F_\tau E R \nu^{-\frac{s}{2}} + \Delta_1 I_{s>2}, \tag{23}
$$

where C is an absolute constant.

### A.1.3    PROOF OF LEMMA 7

**Lemma 9** (Lemma 13 in Fischer & Steinwart (2020)). *Let $(\mathcal{X}, B)$ be a measurable space, $\mathcal{H}$ be a separable RKHS on $\mathcal{X}$ w.r.t. a bounded and measurable kernel $k$, and $\mu$ be a probability distribution on $\mathcal{X}$. Then the following equality is satisfied, for $\nu > 0$,*

$$
\int_{\mathcal{X}} \left\| (T_\nu)^{-1/2} K(x, \cdot) \right\|_{\mathcal{H}}^2 d\mu(x) = N(\nu).
$$

*If, in addition, $M_\alpha < \infty$ is satisfied, then the following inequality is satisfied, for $\nu > 0$ and $\mu$-almost all $x \in \mathcal{X}$,*

$$
\left\| (T_\nu)^{-1/2} K(x, \cdot) \right\|_{\mathcal{H}}^2 \leq M_\alpha^2 \nu^{-\alpha}.
$$

*we also consider $C_x : \mathcal{H} \rightarrow \mathcal{H}$ the integral operator w.r.t. the point measure at $x \in \mathcal{X}$,*

$$
C_x f := f(x) K(x, \cdot) = \langle f, K(x, \cdot) \rangle_{\mathcal{H}} K(x, \cdot),
$$

*And we have the operator norm:*

$$
\left\| (T_\nu)^{-1/2} C_x (T_\nu)^{-1/2} \right\|_{\mathscr{B}(\mathcal{H})} = \left\| (T_\nu)^{-1/2} K(x, \cdot) \right\|_{\mathcal{H}}^2 \leq M_\alpha^2 \nu^\alpha.
$$

**Lemma 10** (Bernstein's Inequality in Caponnetto & De Vito (2007))**.** *Let $(\Omega, B, P)$ be a probability space, $H$ be a separable Hilbert space, and $\xi : \Omega \to H$ be a random variable with*

$$E_P[\|\xi\|_{\mathcal{H}}^2] \leq \sigma^2,$$

$$\|\xi(\omega)\|_{\mathcal{H}} \leq \frac{L}{2}, \quad a.s.$$

*Then, for $\tau \geq 1$ and $n \geq 1$, the following concentration inequality is satisfied*

$$\mathbb{P}^n \left( (\omega_1, \ldots, \omega_n) \in \Omega^n : \left\| \frac{1}{n} \sum_{i=1}^n \xi(\omega_i) - E_P \xi \right\|_{\mathcal{H}}^2 \geq 32 \frac{\tau^2}{n} \left( \sigma^2 + \frac{L^2}{n} \right) \right) \leq 2e^{-\tau}.$$

*Proof of Lemma 7.* Consider the random variable $\xi : X \times \{-1, 1\} \to \mathcal{H}$:

$$\xi(x, y) = (T_\nu)^{-1/2} (y - f_\nu(x)) K(x, \cdot).$$

Denoted by $D$ the empirical measure corresponding to the sample $\{X_i, Y_i\}_{i=1}^n$. It holds that

$$\frac{1}{n} \sum_{i=1}^n (\xi(X_i, U_i) - E_P \xi) = E_D \xi - E \xi$$

$$= (T_\nu)^{-1/2} ((g_Z - T_X f_\nu) - (g - T f_\nu)).$$

By Lemma 9, it holds that

$$\|\xi(x, y)\|_{\mathcal{H}} \leq 2 \left\| (T_\nu)^{-1/2} K(x, \cdot) \right\|_{\mathcal{H}} \leq 2 M_\alpha \nu^{\frac{\alpha}{2}}$$

for $\mu$-almost all $x \in \mathcal{X}$, and it holds that

$$E(\|\xi(x, y)\|_{\mathcal{H}}^2) = \int_{\mathcal{X}} \left\| (T_\nu)^{-1/2} K(x, \cdot) \right\|_{\mathcal{H}}^2 \left[ \sum_{i=1,-1} |i - f_\nu(x)|^2 P(Y = i \mid X = x) \right] d\mu(x)$$

$$\leq 4N(\nu).$$

We let $\sigma^2 = 4N(\nu)$ and $L = 4M_\alpha \nu^{\frac{\alpha}{2}}$ and apply the Bernstein inequality in Lemma 10 to complete the proof. $\qquad \square$

### A.2 UPPER BOUND ON EXCESS RISK

**Theorem 3** ($L^2$-risk upper bound)**.** *Suppose that Assumption 4, 5 and 6 holds, and that $0 < s \leq 2\tau$. Let $\hat{f}_\nu$ be the estimator defined by equation 17. Then by choosing $\nu \asymp n^{\frac{\beta}{s\beta+1}}$, for any fixed $\delta \in (0, 1)$ and any $1 \geq \alpha > \alpha_0$, when $n$ is sufficiently large, with probability at least $1 - \delta$, we have*

$$\left\| \hat{f}_\nu - f_\rho^* \right\|_{L^2}^2 \leq \left( \ln \frac{6}{\delta} \right)^2 C n^{-\frac{s\beta}{s\beta+1}},$$

*where $C$ is a constant independent of $n$ and $\delta$.*

*Proof.* We decomposed $L^2$-risk into the sum of the approximation error and the estimation error as as

$$\|\hat{f}_\nu - f_\rho^*\|_{L^2} \leq \|f_\nu - f_\rho^*\|_{L^2} + \|\hat{f}_\nu - f_\nu\|_{L^2}.$$

Using Lemma 2 for the appoximation error and equation 23 for the estimation error, we have

$$\|\hat{f}_\nu - f_\rho^*\|_{L^2} \leq C \left( \log \frac{2}{\delta} \right) \left( \frac{N(\nu)^{\frac{1}{2}}}{\sqrt{n}} + \frac{2M_\alpha \nu^{\frac{\alpha}{2}}}{n} \right) + F_\tau (E + 1) R \nu^{-\frac{s}{2}} + \Delta_1 I_{s>2}.$$

Choosing $\nu \asymp n^{\frac{\beta}{s\beta+1}}$, we can obtain the following rates:

- By Lemma 1,

$$\frac{N(\nu)^{\frac{1}{2}}}{\sqrt{n}} \asymp \frac{\nu^{\frac{1}{2\beta}}}{\sqrt{n}} = n^{-\frac{1}{2}\frac{s\beta}{s\beta+1}} \tag{24}$$

-

$$F_\tau(E+1)R\nu^{-\frac{s}{2}} \asymp n^{-\frac{1}{2}\frac{s\beta}{s\beta+1}} \tag{25}$$

- Following Theorem 16 in Zhang et al. (2023), for any $s > 2$, we have

$$\Delta_1 \lesssim \left(\ln\frac{6}{\delta}\right) n^{-\frac{1}{2}\frac{s\beta}{s\beta+1}} \tag{26}$$

- Since $\alpha_0 = \frac{1}{\beta}$ for Sobolev RKHS, we can choose some $\alpha$ in $(\alpha_0, s + \frac{2}{\beta})$. It follows that

$$\frac{\nu^{\frac{\alpha}{2}}}{n} \lesssim n^{-\frac{1}{2}\frac{s\beta}{s\beta+1}} \tag{27}$$

Combining the above equation 24, equation 25, equation 26 and equation 27, we complete the proof. $\square$

*Proof of Theorem 2*. The classification excess risk can be rewritten as

$$\begin{aligned}
&\mathcal{E}(\hat{f}) = \mathcal{L}(\hat{f}) - \mathcal{L}^* \\
=&\boldsymbol{E}_X[(1-\eta(X))(\mathbb{I}\{\hat{f}(X) \geq 0\} - \mathbb{I}\{2\eta(X) - 1 \geq 0\}) + \eta(X)(\mathbb{I}\{\hat{f}(X) < 0\} - \mathbb{I}\{2\eta(X) - 1 < 0\}))] \\
=&\boldsymbol{E}_X(|2\eta(X) - 1|\mathbb{I}\{\hat{f}(X)(2\eta(X) - 1) < 0\}) \\
=&\boldsymbol{E}_X(|f_\rho^*(X)|\mathbb{I}\{\hat{f}(X)f_\rho^*(X) < 0\}).
\end{aligned} \tag{28}$$

Based on equation 28 and the classic upper classification upper bound by Devroye et al. (2013), we have:

$$\begin{aligned}
\mathcal{E}(\hat{f}_\nu) &= \boldsymbol{E}_X(|f_\rho^*(X)|\mathbb{I}\{\hat{f}_\nu(X)f_\rho^*(X) < 0\}) \\
&\leq \int_{\mathcal{X}} \left|f_\rho^*(X) - \hat{f}_\nu(X)\right| d\nu(X) \\
&\leq \|\hat{f}_\nu - f_\rho^*\|_{L^2}
\end{aligned}$$

The remaining proof directly follows from Theorem 3. $\square$

### A.3 MINIMAX LOWER BOUND

**Proposition 2** (Theorem 2.5 in Tsybakov (2009)). *Assume that $M \geq 2$ and suppose that $\Theta$ contains elements $\theta_0, \theta_1, \ldots, \theta_M$ and $P_{\theta_0}, P_{\theta_1}, \ldots, P_{\theta_M}$ are the probability measures such that*

*(i) $d(\theta_i, \theta_j) \geq 2s > 0$, $\forall 0 \leq i \leq j \leq M$;*

*(ii) $P_{\theta_j} \ll P_{\theta_0}$, $\forall j = 1, \ldots, M$ and*

$$\frac{1}{M}\sum_{j=1}^M KL(P_{\theta_j}, P_{\theta_0}) \leq a \log M \tag{29}$$

*with $0 < \alpha < 1/8$.*

*Then*

$$\inf_{\hat{\theta}} \sup_{\theta \in \Theta} \mathbb{P}_\theta(d(\hat{\theta}, \theta) > s) \geq \frac{\sqrt{M}}{1 + \sqrt{M}}\left(1 - 2\alpha - \sqrt{\frac{2\alpha}{\log M}}\right) > 0 \tag{30}$$

**Lemma 11** (Varshamov-Gilbert Bound). *Given $m \geq 8$, there exist $M \geq 2^{m/8}$ different elements on $\omega^{(0)}, ..., \omega^{(M)}$ on $\{-1, 1\}^m$ and $\omega^{(0)} = (0, ..., 0)$ such that*

$$\sum_{k=1}^{m} |\omega_k^{(i)} - \omega^{(j)}| \geq \frac{m}{4}, \quad 0 \leq i < j \leq M. \tag{31}$$

**Lemma 12.** *For $r > \frac{d}{2}$, $H^r(\mathcal{X})$ is a separable RKHS with respect to a bound kernel and the corresponding EDR is*

$$\beta = \frac{2r}{d}.$$

Let $u : \mathbb{R}_+ \to \mathbb{R}_+$ be a nonincreasing infinitely differentiable function such that $u = 1$ on $[0, 1/4]$ and $u = 0$ on $[1/2, \infty)$. We can take $u(x) = \left( \int_{1/4}^{1/2} u_1(z) \mathrm{d}z \right)^{-1} \int_x^\infty u_1(z) \mathrm{d}z$ where

$$u_1(x) = \begin{cases} \exp\left\{ -\frac{1}{(1/2-x)(x-1/4)} \right\}, & \text{for } x \in (1/4, 1/2), \\ 0, & \text{otherwise.} \end{cases} \tag{32}$$

Given an integral $q = q(n) \geq 1$, we define the regular grid on $\mathbb{R}^d$ as

$$G_q = \left\{ \left( \frac{2k_1 + 1}{2q}, ..., \frac{2k_d + 1}{2q} \right) : k_i \in \{0, ..., q-1\}, i = 1, ..., d \right\}. \tag{33}$$

We consider the partition $\mathcal{X}_1, ..., \mathcal{X}_{q^d}$ of $[0, 1]^d$ canonically defined using the grid $G_q$. $x \in \mathcal{X}_k$ if $z_k \in G_q$ is the closest point to $x \in [0, 1]^d$. If there exist several points in $G_q$ closest to $x$ we define $x \in \mathcal{X}_k$ if $z_k$ is closest to 0.

**Lemma 13.** $\psi(x) = C_\psi q^{-sr} \sum_{k=1}^{q^d} \phi(q[x - z_k]) \in H^{sr}(\mathcal{X})$, where $\phi(x) = u(\|x\|_2)$.

*Proof of Lemma 13.* By the definition of $\phi$, we have $\phi \in H^{sr}$ for any fixed $sr > 0$. Thus, $\|\phi\|_{H^{sr}}^2$ is bounded.

Denote $\psi_k(x) = \phi(q[x - z_k])$ and $\varphi(x) = \psi_k(x + z_k) = \phi(qx)$. It is easy to find that: $i)$ $\|\psi_k\|_{[H^{sr}]} = \|\varphi\|_{H^{sr}}$ ; $ii)$ $\langle \psi_i, \psi_j \rangle_{H^{sr}} = 0$ for $i \neq j$.

$$\|\psi\|_{H^{sr}}^2 = C_\psi^2 q^{-2sr} \| \sum_{k=1}^{q^d} \psi_k \|_{H^{sr}}^2 \tag{34}$$

$$= C_\psi^2 q^{-2sr} \left( \sum_{k=1}^{q^d} \|\psi_k\|_{H^{sr}}^2 + 2 \sum_{i \neq j} \langle \psi_i, \psi_j \rangle_{H^{sr}}^2 \right) \tag{35}$$

$$= C_\psi^2 q^{d-2sr} \|\varphi\|_{H^{sr}}^2 \tag{36}$$

Denote the Fourier transform of $\phi$ and $\psi$ as $\hat{\phi}$ and $\hat{\psi}$

$$\hat{\varphi}(\xi) = \int_{B(0, \frac{1}{2q})} \phi(qx) e^{-2\pi i \xi x} \mathrm{d}x \tag{37}$$

$$= q^{-d} \int_{B(0, \frac{1}{2})} \phi(y) e^{-2\pi i \frac{\xi}{q} y} \mathrm{d}y \tag{38}$$

$$= q^{-d} \hat{\phi}(\frac{\xi}{q}) \tag{39}$$

Since $u(x)$ is infinitely differentiable function on $[0, 1]^d$, then $\|\phi\|_{H^s}$ is bounded for any fixed $s > 0$. Then

$$\|\varphi\|_{H^{sr}}^2 = q^{-2d} \int_{R^d} |\hat{\phi}(\frac{\xi}{q})|^2 (1 + \|\xi\|_2^2)^s \mathrm{d}\xi \tag{40}$$

$$= q^{-2d} \int_{R^d} |\hat{\phi}(z)|^2 (1 + q^2\|z\|_2^2)^{sr} q^d \mathrm{d}z \tag{41}$$

$$\leq q^{2sr-d} \int_{R^d} |\hat{\phi}(z)|^2 (1 + \|z\|_2^2)^{sr} \mathrm{d}z \tag{42}$$

$$= q^{2sr-d} \|\phi\|_{H^{sr}}^2. \tag{43}$$

Thus, $\|\psi\|_{H^{sr}}^2 \leq C_\psi^2 \|\phi\|_{H^{sr}}^2$. $\qquad\square$

*Proof of Theorem 1.* Denote $m = q^d$, we define $\mathcal{X}_0 = \mathbb{R}^d \setminus \bigcup_{i=1}^m \mathcal{X}_i$. Thus, $\mathcal{X}_0, ..., \mathcal{X}_m$ form a partition of $\mathbb{R}^d$.

Define the hypercube $\mathcal{C} = \{P_\omega : \omega = (\omega_1, ..., \omega_m) \in \{-1, 1\}^m\}$ of probability distribution $P_\omega$ of $(X, Y)$ on $Z = \mathbb{R}^d \times \{0, 1\}$ as follows.

For any $P_\omega$, the marginal distribution of $X$ does not depend on $\omega$ and has a density $\mu$ w.r.t. the Lebesgue measure on $\mathbb{R}^d$ defined in the following way. Denote $v = m^{-1}$. Let $\mu(x) = v/\lambda[B(0, (4q)^{-1})]$ for $x \in B(z, (4q)^{-1})$, $z \in G_q$ and $\mu(x) = 0$ otherwise.

By Lemma 11, there exist $M \geq 2^{m/8}$ different elements on $\omega^{(0)}, ..., \omega^{(M)}$ on $\{-1, 1\}^m$ and $\omega^{(0)} = (0, ..., 0)$. We take

$$f_i(x) = \omega_k^{(i)} \psi(x), \quad x \in \mathcal{X}_k, \quad i \in [M] \tag{44}$$

and $f_i(x) = 0$ for $x \in \mathcal{X}_0$. $f_0 = 0$ for $x \in \mathcal{X}$. We will assume that $C_\psi \leq 1$ to ensure that $\eta_i(x) = (1 + f_i(x))/2$ take values in $[0, 1]$.

By Lemma 13, we have $f_i \in H^{sr} = [\mathcal{H}]^s$, $i = 0, ..., M$. Since $\eta_i(x) = (1 + f_i(x))/2$, we have

$$KL(\rho_i^n, \rho_0^n) = nKL(\rho_i, \rho_0) \tag{45}$$

$$= n \sum_{k=1}^m \int_{\mathcal{X}_k} \left( \frac{1 + \omega_k^{(i)}\psi(x)}{2} \ln(1 + \omega_k^{(i)}\psi(x)) + \frac{1 - \omega_k^{(i)}\psi(x)}{2} \ln(1 - \omega_k^{(i)}\psi(x)) \right) \mu(x)\mathrm{d}x \tag{46}$$

$$\leq \frac{1}{2} nm \int_{B(x_0, (4q)^{-1})} \frac{v}{\lambda[B(0, (4q)^{-1})]} \left( \ln(1 - \psi^2(x)) + \psi(x) \ln\left(\frac{1 + \psi(x)}{1 - \psi(x)}\right) \right) \mathrm{d}x \tag{47}$$

$$\leq \frac{1}{2} nm \int_{B(x_0, (4q)^{-1})} \frac{v}{\lambda[B(0, (4q)^{-1})]} \left( \psi(x) \ln\left(\frac{1 + \psi(x)}{1 - \psi(x)}\right) \right) \mathrm{d}x \tag{48}$$

$$\leq Cnmvq^{-2sr} \tag{49}$$

The last inequality is because with sufficiently large $n$, $\psi(x) = C_\psi q^{-sr} < 1/2$ for $x \in B(x_0, (4q)^{-1})$. To satisfy the second condition of Proposition 2, we need $C_\psi nmvq^{-2sr} \leq a\frac{m}{8}\ln 2$. We can take $nvq^{-2sr} = \Theta(1)$. Thus, $q = n^{1/(2sr+d)}$.

For the first condition, we have

$$d(f_i, f_j) = \mathbb{E}_x[|f_i(x)|1_{f_i(x)f_j(x)<0}] \tag{50}$$

$$= \mathbb{E}_x \left[ \sum_{k=1}^m \psi(x)1_{\sigma_k^{(i)}\sigma_k^{(j)}<0}1_{x \in \mathcal{X}_k} \right] \tag{51}$$

$$\geq \frac{C_\psi q^{-sr}mv}{4} \tag{52}$$

$$= Cn^{-sr/(2sr+d)} \tag{53}$$

for some constant $C > 0$. By Proposition 2, we have the minimax rate $n^{-sr/(2sr+d)}$. Since $r = \beta d/2$, we have the minimax rate $n^{-\frac{s\beta}{2(s\beta+1)}}$. $\qquad\square$

### A.4 EMBEDDING INDEX OF SOBOLEV AND DOT-PRODUCT KERNELS

#### A.4.1 SOBOLEV KERNEL

The interpolation space of $H^r(\mathcal{X})$ under Lebesgue measure is given by

$$[H^r(\mathcal{X})]^s = H^{rs}(\mathcal{X}). \tag{54}$$

By the embedding theorem of (fractional) Sobolev space (Theorem 4.27 in Adams & Fournier (2003)), letting $\theta = r - \frac{d}{2} > 0$, we have

$$H^r(\mathcal{X}) \hookrightarrow C^{0,\theta}(\mathcal{X}) \hookrightarrow L^\infty(\mathcal{X}), \quad \theta = r - \frac{d}{2}.$$

Combined with equation 54, for a Sobolev RKHS $\mathcal{H} = H^r(\mathcal{X}), r > \frac{d}{2}$ and any $\alpha > \frac{1}{\beta} = \frac{d}{2r}$, we have

$$[H^r(\mathcal{X})]^\alpha = H^{r\alpha}(\mathcal{X}) \hookrightarrow C^{0,\theta}(\mathcal{X}) \hookrightarrow L^\infty(\mathcal{X}).$$

Therefore $\alpha_0 = \frac{1}{\beta}$ for the embedding index of a Sobolev RKHS.

#### A.4.2 DOT-PRODUCT KERNEL

Let $k$ be a dot-product kernel on $X = S^d$, the unit sphere in $R^{d+1}$, and $\mu = \sigma$ be the uniform measure on $S^d$. Then, it is well-known that $k$ can be decomposed as

$$k(x,y) = \sum_{n=0}^\infty \mu_n \sum_{l=1}^{a_n} Y_{n,l}(x) Y_{n,l}(y), \tag{55}$$

where $\{Y_{n,l}\}$ is a set of orthonormal basis of $L^2\left(S^d, \sigma\right)$ called the spherical harmonics. $a_n$ is multiplicity and satisfies

$$a_n := \binom{n+d}{n} - \binom{n-2+d}{n-2}$$

We let $\mu_n \asymp n^{-d\beta}$ for some $\beta > 1$ and also we have $a_n \asymp n^{d-1}$ and $\sum_{i=1}^n a_i \asymp n^d$, we prove embedding index $\alpha_0 = \frac{1}{\beta}$.

*Proof.* By the Theorem 9 in Fischer & Steinwart (2020), we only need to prove that for any $\alpha > \frac{1}{\beta}$, $\sum_{n=0}^\infty \mu_n^\alpha \sum_{l=1}^{a_n} Y_{n,l}(x)^2 < \infty$.

$$\sum_{n=0}^\infty \mu_n^\alpha \sum_{l=1}^{a_n} Y_{n,l}(x)^2 \leq \sum_{n=0}^\infty \mu_n^\alpha a_n$$

$$\leq \sum_{n=0}^\infty C n^{d-1} n^{-\alpha d\beta}$$

$$= C \sum_{n=0}^\infty n^{-1-d(\alpha\beta-1)} < \infty$$

$\square$

## B DETAILED DISCUSSION FOR $f$ WITH COMPLEX STRUCTURES

In this section, we illustrate some $f^*$ with the complex structure and analyze the feasibility of our theories and method on these cases:

**Mixed smoothness:** Suppose $f^* = \sum_{j=1}^{\infty} f_j e_j(x)$ and $\{f_j\}_{j \in N}$ has different different decay rates. A simple example is the two-smoothness case, where $\{f_j\}_{j \in N}$ has two different decay rates, where $f_j^2 = j^{-s_1\beta-1}$ when $j$ is odd and $f_j^2 = j^{-s_2\beta-1}$ when $j$ is even ($s_1 > s_2$). For the kernel $K$ with EDR $\beta$, by the definition of the interpolation space, we have

$$\|f^*\|_{[\mathcal{H}]^s} = \sum_{j=1}^{\infty} \frac{f_j^2}{\lambda_j^s} = \sum_{\{j:j \text{ is odd}\}} j^{s\beta-s_1\beta-1} + \sum_{\{j:j \text{ is even}\}} j^{s\beta-s_2\beta-1} \tag{56}$$

Thus, for $s < s_2$, we have $\|f^*\|_{[\mathcal{H}]^s}$ is bounded, meaning that $f_\rho^* \in [\mathcal{H}]^s$ where $s$ can be arbitrary close to $s_2$. In this case, our theory can still be applied to find the generalization ability of the kernel classifiers ($n^{-\frac{s_2\beta}{s_2\beta+1}}$). This can be also applied to multi-smoothness cases.

However, in this case, Truncation Estimation, introduced in Section 5, is inaccurate. With a sufficient sample size, Truncation Estimation will find the smoothness $s$ between $s_1$ and $s_2$ for the two-smoothness case while we need to find $s = s_2$. In this case, the method can be improved by performing linear regression on top of $\tilde{p}_j = \sup_{k \geq j} p_j$ even though this improvement method tends to underestimate $s$. We will find new $s$ estimation methods for general situations in the near future.

**Sobolev space of low intrinsic dimensionality**   There is a popular assumption on the real data, called manifold assumption, assuming that $f^*$ is supported on a submanifold. More specifically, for $x \in \Omega \subset \mathbb{R}^D$, they assume that $f^*$ belongs to the space of the low intrinsic dimensionality $d$ and $d < D$. In this case, Hamm & Steinwart (2021); Ding et al. (2023) have come up with some definitions of the low intrinsic dimension assumption:

**Assumption 7** (Low intrinsic dimension). *There exist positive constants $c_1$ and $d \leq D$ such that for all $\delta \in (0,1)$, we have*

$$\mathcal{N}_{l_\infty^D}(\delta, \Omega) \leq c\delta^{-d} \tag{57}$$

*where $l_\infty^D$ is the $\mathbb{R}^D$ space equipped with $l_\infty$ norm and $\mathcal{N}_{l_\infty^D}(\delta, \Omega)$ is the covering number.*

On the Sobolev space with smoothness $r$ With the assumption of low intrinsic dimension, Hamm & Steinwart (2021); Ding et al. (2023) improved their results from $n^{-\frac{2r}{2r+D}}$ to $n^{-\frac{2r}{2r+d}}$ (regression problems). Though our theories can not solve this case, we believe that the technology can be applied to classification problems and this will be our future work.

**Well-separated data**   The well-separated assumption is another popular assumption on the real data (like MNIST, CIFAR-10, and so on) since the testing accuracy of some neural network models is near $100\%$. The well-separated assumption, in our settings, means that $f^*(x) \in \{1, -1\}$, violating the continuity of $f^*$. However, we can use a continuous function to approximate such a discontinuous function. For example, $x \in [0,1]$ and $f^*(x) = 1$ if $x > 1/2$ and $f^*(x) = -1$ if $x \leq 1/2$ (two regions). Then we can use an infinitely differentiable function ($s = \infty$) to approximate $f^*$ and thus the estimator finds out that the function is arbitrarily smooth. This idea can be extended to the cases with finite regions.

However, for the real data, normally with a super large dimension, the number of regions may depend on the dimension. In this situation, our theories need more effort to explain the generalization ability of the kernel classifier (like extending our theories to the high dimensional settings).