# OpenReview forum: "The optimality of kernel classifiers in Sobolev space"
_ICLR.cc/2024/Conference — ICLR 2024 poster_

### Official Review · Reviewer_atim · 2023-10-31

**Soundness:** 3 good
**Presentation:** 2 fair
**Contribution:** 3 good
**Rating:** 5
**Confidence:** 2

**Summary:**

In this paper, the authors investigate the statistical performance of kernel classifier, and establish, under rather standard assumptions in Theorem 1 and 2, respectively, upper and lower bounds on the classification excess risk (over Sobolev spaces), showing the optimally of the proposed classifier.

The proposed theoretical analysis is then applied to (deep) neural network (NN) models, to establish some generalization error bound of a fine-tuned NN model in the over-parameterization regime.

An intuitive approach is also proposed in Section 5 to estimate the key smoothness parameter with respect to the kernel.

**Strengths:**

This paper focuses on the fundamental problem of kernel learning and, if I understand correctly, improves some previous efforts such as Kerkyacharian & Picard (1992).

The obtained results (Theorem 1 and 2) are rather strong, in that Theorem 1 and 2 together show the minimax rate optimality of the proposed kernel spectral classifier.

The authors establish an interesting connection between the proposed theoretical analysis to NTK and deep neural networks in Corollary 1.

**Weaknesses:**

I am not an expert in the theory of kernel learning and I personally find this paper a bit difficult to digest. I assume some other audiences from the (theoretical) AI/ML community may have the same impression.
I think this paper can be improved in terms of presentation. See my detailed comments below.

**Questions:**

I the following questions and/or comments for the authors:

1. Page 3: "Therefore, the framework of spectra algorithm" -> spectral algorithm. Similarly for the first sentence in Page 5.
2. Page 5: "bounded domain with smooth boundary $\mathcal{X} \subseteq R^d$" should be $\mathcal{X} \subseteq \mathbb{R}^d$ here?
3. I do not understand Assumption 3 and the notations therein, could the authors clarify this?
4. The proof of Theorem 1-2, Corollary 2, and the appendix should be re-organized for better readability.
5. I do not understand the statement of "width m and the sample size n are sufficiently large" in Corollary 1: should Corollary 1 be understood as asymptotic statements as $m,n \to \infty$? I am a bit confused since in many DNN and NTK literature, one must have $m \gg n$ for the NTK to accurately approximate the DNN behavior. Also, can the results in Corollary 1 be compared to existing results in NTK kernel literature?
6. I do not understand the connection between the min kernel discussed above Equation (17) and the NTK.

---

> ### Author Response · Authors · 2023-11-15
> **Thank you for your review**
>
> Thank you for your nice summary of our paper and your valuable feedback.
>
> For Questions 1,2 and 4, we will correct them and re-organize the proofs in the final version.
>
> -----------------
>
> Question 3: I do not understand Assumption 3 and the notations therein, could the authors clarify this?
>
> Answer: Assumption 3 is corresponding to how a RKHS is compactly embedded into $L^{\infty}$-space. This directly implies that all the functions in $[\mathcal{H}]^{\alpha}$ are $\mu$-a.e bounded for $\alpha>\alpha_0$.
>
> -----------------
>
> Answer to Question 5: Generally speaking, [1] studying the neural network with NTK can be divided into the following steps. In the first step, we want to verify the time-varying Neural network kernel function uniformly converges to NTK kernel, where a large enough wide $m$ corresponding to sample size $n$ is needed,  such as $m \geq \operatorname{poly}\left(n, || y ||, \ln (1 / \delta)\right)$. Therefore, if the training process of the neural network uniformly converges to NTK kernel regression, we need the spectral properties of NTK, in other words, the eigenvalue decay rate. The last step is to analyze the kernel regression, where the spectral algorithm is considered because we can study not only ridge regression but also gradient descent and other training process.
>
> NTK kernel provides a perspective of kernel methods to study the neural network. Although it is based on a wide neural network in which we need a large enough $m$ corresponding to $n$, we can study the generalization properties and understand the benefits of over-parameterization in the neural network.
>
> -----------------
>
> Question 6:  I do not understand the connection between the min kernel discussed above Equation (17) and the NTK.
>
> Answer:  The min kernel is an example to illustrate how to estimate $s$. The reason why we use the min kernel is that we know the eigenvalue and explicit eigenfunction of the min kernel on $\mathbb{R}$. Therefore, we can calculate the theoretical smoothness $s$ for given $f^*$ and the optimal rate. Then we compare them with our numeric results. After showing the way to estimate $s$ by using the min kernel, we then estimate $s$ of the real data with respect to the NTK kernels.
>
> [1]Yicheng Li, Zixiong Yu, Guhan Chen, and Qian Lin. Statistical optimality of deep wide neural networks. arXiv preprint arXiv:2305.02657, 2023.

---

> > ### Comment · Reviewer_atim · 2023-11-21
> >
> > I thank the authors for their reply.
> >
> > I would like to insist on my (previous) Question 5: I am still a bit confused, so just to make sure if I understand correctly, for the proposed Corollary 1 (and possibly some other results) to hold, there is in fact some condition on the network width $m$, e.g., should be at least in poly of $n$. If this is the case, I believe this should be stated in an explicit fashion in the paper. For now, the condition/assumption seems a bit ambiguous.
> >
> > Also, kindly note that the authors are **allowed** to revise and update revised version of the paper during the rebuttal of ICLR 2024.

---

> > > ### Author Response · Authors · 2023-11-21
> > > **Thank you for your kind reminder**
> > >
> > > Thank you for your comment and reminder.
> > >
> > > Your understanding is correct. We have updated the revised version of the paper and included the clear condition of the width for Proposition 1 $m \geq poly(n, \lambda_{min}^{-1},  \ln (1 / \delta), \ln(1/\epsilon) )$, where $\lambda_{min}$ is the minimum eigenvalue of the kernel matrix, and for Corollary 1  $m \geq \text{poly}\left(n, \lambda_{min}^{-1}, \ln (1 / \delta) \right)$.

---

### Official Review · Reviewer_3CL4 · 2023-11-06

**Soundness:** 3 good
**Presentation:** 3 good
**Contribution:** 3 good
**Rating:** 6
**Confidence:** 4

**Summary:**

The paper shows that a variety of kernel regression methods known as spectral algorithms achieve optimal convergence rates for binary classification when the conditional probability function $\eta$ stems from a Sobolev space. The authors also apply this theory to neural tangent kernels. Finally, the paper provides a simple (but not theoretically analyzed) method for estimating the smoothness of the conditional probability function $\eta$ from a data set, which is applied to standard image classification data sets.

**Strengths:**

The paper has a relatively clear message, and I think the paper is relatively well-written. I guess that the upper bound on the convergence rate in Theorem 2 should be a relatively direct corollary from Fischer and Steinwart (2020) in the special case of ridge regularization, but the authors formulate a more general result for spectral algorithms that also holds for gradient flow. The lower bound in Theorem 1 nicely complements the upper bound. While the square loss is not very common for classification, it is still noteworthy that optimal rates can be achieved with any loss.

**Weaknesses:**

The application to the NTK looks nice, but I have some concerns that Assumption 3 cannot be verified for a good $\alpha_0$, see the major comments below.

While the smoothness assumption on $f^*$ seems less natural to me in the classification setting, the authors provide interesting experiments that appear to show that this assumption is appropriate. However, I have doubts that the proposed (and not theoretically justified) estimator can really estimate what it is supposed to estimate, see the major comments below.

Overall, the aforementioned concerns hold me back from recommending acceptance at the moment, but I am ready to adjust my score if these concerns can be addressed.

**Questions:**

**Questions**:
(1) Does ridge regularization also qualify as a spectral algorithm? And what about gradient descent?

(2) Could the results be generalized to multi-class classification? I am also wondering about other loss functions, but the Brier score should be trivial and the log-loss should require additional modifications.

**Major comments**:
(3) Before Corollary 1: While NTKs are dot-product kernels on the sphere, they are not in general dot-product kernels outside of the sphere. For example, from Eq. (15) we have $K_{ntk}(0, 0) = 1$, which means that $K_{ntk}$ cannot be a dot-product kernel. On the other hand, I also don't see why you would need it to be a dot-product kernel for Corollary 1.

(4) In Theorem 2, you seem to implicitly assume that $\alpha_0 = 1/\beta$, see the text before Eq. (14) in the appendix. Please state this in the theorem. Moreover, it is not clear to me that this would also hold in the setting of Corollary 1 for general domains. In the special case of the sphere, this could be fixed at least for very similar network architectures because then the RKHS of the NTK is known to be a Sobolev space, see for example the proof of Theorem G.5 in https://arxiv.org/abs/2305.14077

(5) I am not fully convinced by the proposed smoothness estimator. For example, assume that the target function $f^*: \mathbb{R}^2 \to \mathbb{R}$ has different degrees $s_1$ and $s_2$ of (Sobolev) smoothness in dimensions $1$ and $2$. Then, I would expect the eigenvalues to decay at different rates depending on the direction that the eigenfunctions are more refined in, such that the eigenvalues in a plot as in Figure 1 (a) wouldn't lie on a single line but in between two lines corresponding to $s_1$ and $s_2$. Linear regression would then estimate a smoothness between $s_1$ and $s_2$, but in terms of interpolation spaces, the function would still only lie in the interpolation space with the lower smoothness. (This could be checked by just plotting the $p_j$ on the image data sets.)

Perhaps this issue could be "fixed" by performing linear regression on top of $\tilde p_j := \sup_{k \geq j} p_k$, which would try to filter out the "upper line" in the plot, but also potentially it could be more susceptible to noise. (Judging from the plots, perhaps it could be a good heuristic to automatically set the threshold in a way that only those $p_j$ are included that are larger than the maximum in the right half of the plot.) Another idea for the estimator would be to stay closer to the definition of interpolation spaces and work with (approximations of) the interpolation space norms of the regression function, maybe something like finding an "elbow" in $s \mapsto y^\top (K + \lambda I)^{-1} K^s (K + \lambda I)^{-1} y$.

Besides the mixed smoothness case, another important case is the "manifold assumption" case where $\mu$ is supported on a submanifold, and which is a popular assumption for image data sets. I am wondering whether the estimator can be trusted in this case.
Another potential test case is what happens if the classes are well-separated and non-noisy, i.e., with $\eta(x) \in \{1, -1, \text{undefined}\}$. This is arguably close to the situation for MNIST. Can the estimator find out that the function is arbitrarily smooth?
In general, it would be interesting to see whether the rates predicted by the theory and the simple estimation method hold up in a practical setting.
(I'm brainstorming a bit here, it is not necessary to try everything in order to address my comment.)

**Minor comments**:
- p. 1: Use \operatorname{sign} or something like this to set the sign operator in non-italic font. (same for later)
- "trained via the gradient flow" -> "trained via gradient flow"
- Sec 1.1 i) "bounded by $...$" there should be a constant $C$ or an O-notation in there.
- Please don't use all-caps for author names in citations (STEINWART & SCOVEL, 2007).
- p. 3: "Recently, Deep" -> "Recently, deep"
- "gained incredible success" -> "achieved incredible success" or so?
- "Another reason we choose" -> "Another reason why we choose"?
- "spectra algorithm" - should this be "spectral algorithms"? (also further below)
- Same paragraph (last paragraph of Sec. 1): When you say "Another reason ... NTK kernel ..." it seems that this is essentially the same reason as before, just formulated in more detail.
- Sec 2.3: Mention that it is gradient flow on the least-squares loss?
- Assumption 3 could be formulated more clearly: $\alpha_0$ always exists because $\alpha = 1$ is always admissible by the bounded kernel assumption. Moreover, $\alpha_0 > 0$ trivially because of the way that $\alpha_0$ is defined. So here, I think Assumption 3 should be a definition (something like "We define the embedding index $\alpha_0$ as ..."), unless you want to require $\alpha_0 < 1$.
- p. 6: "minimax optimality of kernel classifier" -> "minimax optimality of kernel classifiers"
- p. 6: "source condition 1" -> "source condition (Assumption 1)"
- "Assumption 1, 2, and 3" -> "Assumptions 1, 2, and 3"?
- In Theorem 2: "Assumption 1, 2, and3" -> whitespace is missing
- "of the neural network classifier" -> "of neural network classifiers"
- Eq. (14): The last layer should use the matrix $W^L$, not $W^{(L, p)}$. (The latter notation seems to be a remnant of the mirrored initialization notation, which you do not adopt in the other parts of Eq. (14).)
- The "mirrored initialization" seems to be already older and known as "antisymmetric initialization trick": http://proceedings.mlr.press/v107/zhang20a.html
- The citation Hui (2020) is missing the second author M. Belkin.
- The sentence before Proposition 1 is not ended by a period.
- Before corollary 1: I think the eigenvalue decay shouldn't be $\beta = d(d-1)$, maybe you meant $\beta = d/(d-1)$?
- p. 7: "ground true function" -> "ground-truth function"  (also in the conclusion)
- p. 8, paragraph "Estimation of $s$ in regression": There is nothing sensible after the $\in$ symbol
- Related work: The authors could (briefly) mention/discuss other classes of assumptions that can be made in the classification setting, see e.g. https://projecteuclid.org/journals/electronic-journal-of-statistics/volume-12/issue-1/Improved-classification-rates-under-refined-margin-conditions/10.1214/18-EJS1406.full

**Summary of discussion**:
I missed an assumption in the NTK part, now I am not concerned about its correctness anymore. The smoothness estimator is still not reliable in general, which means that the experiments cannot fully support the appropriateness of the Sobolev assumption, but the authors acknowledge the limitations. In conclusion, I am changing my score from 5 to 6.

---

> ### Author Response · Authors · 2023-11-15
> **Thank you for your review**
>
> Thank you for your review and your thoughtful and comprehensive comments. Question (5) gives us a lot of inspiration and we will consider these problems as our future work (including the more accurate estimation of $s$, combining the manifold settings in our theories and considering the well-separated case).
>
> ---------------------
>
> Answer to Question (1): According to [1], ridge regularization and gradient descent algorithms are both spectral algorithms. Other spectral algorithms consist of Spectral cut-off and continuous version gradient descent (gradient flow) and so on.
>
> ---------------------
>
> Answer to Question (2): We think kernel methods with squared loss can be extended to multi-class classification for we can consider our label a $(s-1)$-dimensional vector, where $s$ is the number of classes. However, the minimax rate for multi-class classification is not clear by far, we only know that [2] obtained a minimax lower bound for VC-class, but it is unknown for RKHS. I think it is a good question for further work.
>
> For the loss function, we think that there is no clear evidence that cross-entropy loss can work better than the squared loss function. In contrast, analyzing the classification upper bound of logistic regression in statistics [3] requires adding some additional unnatural conditions.
>
> ---------------------
>
> Answer to Question (3): We apologize for the mistakes in the formulas of the neural network and the NTK. We consider the neural network without bias on all layers on $\mathbb{S}^{d-1}$. More specifically, we consider the neural network with $x，x'\in\mathbb{S}^{d-1}$:
>     \begin{align}
>     h^{1}(x) & = \sqrt{\frac{2}{m}} \sigma (Ax),\quad h^{l}(x)  = \sqrt{\frac{2}{m}}\sigma(W^{l-1}h^{l-1}(x)),\quad l=2,...,L \\\\
>     f(x;\theta) & = W^{L}h^{L}(x)\quad \text{and} \quad \hat{y} = \operatorname{sign}(f(x;\theta)),
>   \end{align}
>   where $L\geq 2$, and the corresponding NTK
>     \begin{equation}
> K_{ntk}(x,x') = \sum_{r=0}^L \kappa^{(r)}_1(\bar{u}) \prod_s^{L-1}  \kappa_0(\kappa^{(s)}_1(\bar{u}))
>   \end{equation}
>
> where $\bar{u} = \langle x,x'\rangle $. In this case, the NTK is the dot-product kernel. Assuming $x\in\mathbb{S}^{d-1}$, the embedding index of inner product kernels is $\alpha_0=1/\beta$. We will correct the formulas of the neural network and the NTK in the text and include more details on the embedding index in the appendix.
>
> ---------------------
>
> Answer to Question (4): For Sobolev kernels and dot-product kernels on the sphere, we can prove that $\alpha_0 = \frac{1}{\beta}$. That means $\alpha_0 = \frac{1}{\beta}$ for the NTK on the sphere. We will give more details in the appendix.
>
> ---------------------
>
> Answer to Question (5):  The misspecified spectral algorithms (assuming $f^*_{\rho}\in[\mathcal{H}]^s$) have been studied since 2009 (e.g., [4,5,6,7,8]). However, to the best of our knowledge, there is barely any result on the estimation of the smoothness $s$. This paper is the first to propose the $s$ estimation problem and the more accurate $s$ estimation will be our future works.
>
> There are three cases you have mentioned, the mixed smoothness case, the manifold case and the well-separated case. For the mixed smoothness case,  suppose $f^*_\rho= \sum_{j=1}^{\infty}f_je_j(x)$ and $f_j^2 = j^{-s_1\beta-1}$ if $j$ is odd and $f_j^2 = j^{-s_2\beta-1}$ if $j$ is even. Let's say $s_1>s_2$. Then for the kernel $K$ with eigenvalue decay rate (EDR) $\beta$, $f^*_\rho\in[\mathcal{H}]^{s_2}$ instead of $[\mathcal{H}]^{s_1}$ or $[\mathcal{H}]^{s}$ for some $s\in(s_2,s_1)$, meaning that the generalization error only depends on the 'upper line'. Thus, our theorem shows that we should estimate the slope of the 'upper line'. As for the $s$ estimation, in this case, we can estimate $s$ like your idea, performing linear regression on top of $\hat{p}_j$. The estimation indeed becomes more susceptible to noise in this case. However, our experiments show that our estimation becomes more accurate as the sample size $n$ increases. We will try to find more accurate $s$ estimation methods in the near future.
>
> We have to admit that our theories need more effort to support the rest two cases. For the manifold case, so far we can not put the information on the submanifold into our theories. However, we speculate that compared with the data with $\mu$ supported on the original manifold, the data with $\mu$ supported on the submanifold has larger $s$ since the information on $f^*$ is more concentrated, meaning that the projection of $f^*$ might be on the top eigenfunction (the low-frequency functions). Thus, the estimation of $s$ might be used in the manifold case.

---

> > ### Author Response · Authors · 2023-11-15
> >
> > For the well-separated case, this setting may violate the continuity of the ground-true function since $f^*(x)\in\{1,-1\}$. However, we can use a continuous function to approximate such a discontinuous function and this needs more conditions on $f^*$, like the number of the regions. We consider an easy case: $x\in[0,1]$ and $f^*(x)=1$ if $x>1/2$ and $f^*(x)=-1$ if $x\leq 1/2$ (two regions). Then we can use an infinitely differentiable function ($s=\infty$) to approximate $f^*$ and thus the estimator finds out that the function is arbitrarily smooth. However, if the number of the regions is super large (depending on the dimension $d$), then our theories may need more effort to explain this situation.
> >
> >
> > ---------------------
> > Reference:
> >
> > [1] L Lo Gerfo, Lorenzo Rosasco, Francesca Odone, E De Vito, and Alessandro Verri. Spectral algorithms for supervised learning. Neural Computation, 20(7):1873–1897, 2008.
> >
> > [2] Felix Abramovich, Vadim Grinshtein, and Tomer Levy. Multiclass classification by sparse multinomial logistic regression. IEEE Transactions on Information Theory, 67(7):4637–4646, 2021.
> >
> > [3] Felix Abramovich and Vadim Grinshtein. High-dimensional classification by sparse logistic regression. IEEE Transactions on Information Theory, 65(5):3068–3079, 2018.
> >
> > [4]  Ingo Steinwart, Don R Hush, Clint Scovel, et al. Optimal rates for regularized least squares regression. In COLT, pages 79–93, 2009
> >
> > [5] Lee H Dicker, Dean P Foster, and Daniel Hsu. Kernel ridge vs. principal component regression: Minimax bounds and the qualification of regularization operators. Electronic Journal of Statistics, 11(1):1022–1047, 2017.
> >
> > [6] Loucas Pillaud-Vivien, Alessandro Rudi, and Francis R. Bach. Statistical optimality of stochastic gradient descent on hard learning problems through multiple passes. ArXiv, abs/1805.10074, 2018.
> >
> > [7] Simon Fischer and Ingo Steinwart. Sobolev norm learning rates for regularized least squares algorithms. The Journal of Machine Learning Research, 21(1):8464–8501, 2020.
> >
> > [8] Haobo Zhang, Yicheng Li, Weihao Lu, and Qian Lin. On the optimality of misspecified kernel ridge regression. arXiv preprint arXiv:2305.07241, 2023.

---

> > ### Comment · Reviewer_3CL4 · 2023-11-16
> > **Response to the authors**
> >
> > Thank you for the detailed response.
> >
> > (1): Great, it would be helpful to mention this in the paper if you didn't already do so.
> >
> > (2): Thanks for your thoughts.
> >
> > (3): Thanks for the clarification, I overlooked the assumption on the unit sphere. (It is probably better to place it in the Theorem, Theorems can be very misleading if you have to read the whole paper to find out if there is an assumption somewhere.) It is good that you will provide some details in the appendix.
> > The equations could potentially be simplified by using $h^0(x) = x$ as the starting point of the recursion instead of $h^1$.
> > (Maybe it would help to mention that the NTK-RKHS is a Sobolev space only for $L \geq 2$ in the bias-free case.)
> >
> > (4): Thank you. I know that this holds in the Sobolev case. I am a bit surprised that this would hold for general dot-product kernels, though it is definitely plausible for the ReLU NTK with $L=1$ as well.
> >
> > (5): How do you plan to adapt Section 5 in light of the discussed limitations? If the limitations are clearly stated, I would be willing to raise my score to 6. (It is possible to upload a revised version of the paper for the rebuttal.)
> >
> > While thinking about the submanifold case, I also noticed that it is not clear to me from reading Section 5 how exactly your estimator works. Is it just the equation $s = \frac{2r-1}{\beta}$? Why is the numerator $2r-1$ instead of $r$? Is $\beta$ taken from theory or estimated from the data as well?
> >
> > Regarding the submanifold case, following the Sobolev trace theorem, restricting a Sobolev function to a submanifold actually decreases the absolute smoothness $r$, and it should also decrease the relative smoothness $s$. Here is a back-of-the-envelope calculation: Suppose your input space has dimension $D$ but the submanifold has dimension $d < D$, and that your target function satisfies $f^* \in H^a(\mathbb{R}^D)$. By the Sobolev trace theorem, the restriction of $f^*$ to the $d$-dimensional submanifold should typically have Sobolev smoothness $a - (D - d)/2$, which corresponds to an eigenvalue decay rate of $r = (a - (D - d)/2)/(d/2)$. Regardless of how your estimator works, the resulting estimate would probably be affected by the unknown dimension $d$ of the submanifold.

---

> > > ### Author Response · Authors · 2023-11-17
> > > **Thank you for your response**
> > >
> > > Thank you for the detailed response.
> > >
> > > (1): Thank you for reminding us. We have mentioned it on the top of page 5.
> > >
> > > (3): Thank you for your suggestion. We have revised them on pages 6 and 7. And we mention the NTK-RKHS is a Sobolev space before Corollary 1.
> > >
> > > (5): Due to the page limitation, we have mentioned the limitations at the end of Section 5. We give a more detailed discussion of the limitations in the new section (Section B) of the appendix.
> > >
> > > -------
> > >
> > > $s$ estimation: Suppose $f_j=j^{-r}$ and $\lambda_j=j^{-\beta}$. By the definition of the norm of the interpolation space,
> > > \begin{equation}
> > > \sum_{j=1}^{\infty} \frac{f_j^2}{\lambda_j^s} = \sum_{j=1}^{\infty} \frac{j^{-2r}}{j^{-s\beta}} = \sum_{j=1}^{\infty} j^{s\beta-2r}<\infty
> > > \end{equation}
> > > That means if $||f^*||_{[\mathcal{H}]^s}<\infty$, we need $s\beta-2r< -1$, i.e., $s<\frac{2r-1}{\beta}$. $s$ can be arbitrarily close to $\frac{2r-1}{\beta}$.
> > >
> > > For the simulation, $\beta$ is taken from theory since it can be exactly calculated. For the real data, since $d$ is large and the $\beta$ is near 1 no matter from theory or the estimation with data. Thus, $\beta$ is taken from theory in our paper.
> > >
> > > --------
> > >
> > > Thank you for the sharing on the Sobolev trace theorem. Combined with your detailed calculation, we will try to solve the case with the submanifold assumption [1,2] and consider it as our future work.
> > >
> > > Reference:
> > >
> > > [1] Adaptive learning rates for support vector machines working on data with low intrinsic dimension.
> > >
> > > [2] Liang Ding, Tianyang Hu, Jiahang Jiang, Donghao Li, Wenjia Wang, and Yuan Yao. Random smoothing regularization in kernel gradient descent learning. arXiv preprint arXiv:2305.03531, 2023.

---

> > > > ### Comment · Reviewer_3CL4 · 2023-11-17
> > > > **Response to authors**
> > > >
> > > > Thank you for the explanation. I will raise my score to 6. I would suggest to define the smoothness estimator somewhere more explicitly than just implicitly in the min-kernel example.

---

### Official Review · Reviewer_yGYg · 2023-11-07

**Soundness:** 3 good
**Presentation:** 3 good
**Contribution:** 3 good
**Rating:** 5
**Confidence:** 3

**Summary:**

This work shows the minimax optimality of kernel classifiers in Sobolev spaces and extends the theory to neural network classifiers through the connection between neural networks and neural tangent kernels. The paper also shows the error rate depends on the data smoothness. Thus the paper proposes a practical approach to estimate such data smoothness, helping us understand how hard it will be to model the data accurately.

**Strengths:**

* Studying the optimality of kernel classifiers is a fundamental problem in machine learning.
* The theory in this paper has the potential to guide the practice of kernel learning and neural networks.

**Weaknesses:**

The primary concern regarding this paper is that the established minimax optimality for kernel classifiers relies on the gradient flow algorithm, which is mainly based on the L2 loss and is not commonly used in practical applications of building kernel classifiers. While the minimax rate is established, its optimality is only proven in an asymptotic sense, leaving a considerable gap between theory and practical usage. This approach is natural in the existing work when building the minimax rate for kernel classification, but not for the kernel classification. The paper's contribution would be more significant if the theoretical framework were based on widely used classifiers, such as SVM and logistic regression.

A similar issue arises with the application of kernel classifier theory to neural networks, particularly those employing the L2 loss for fitting. Despite the L2 loss being effective in various neural network scenarios, the paper would benefit from incorporating theories that utilize loss functions more prevalent in practice.

An additional point of concern is that the paper's main theoretical underpinnings are based on unpublished works, specifically referenced as https://arxiv.org/abs/2305.07241. The validity and proofs presented in that work have not been verified in this review.

**Questions:**

After estimating the smoothing parameter, can we further gain some insight into the optimal steps in the gradient flow algorithms? It would be helpful if some numerical studies can be performed to support this.

---

> ### Author Response · Authors · 2023-11-15
> **Thank you for your review**
>
> Thank you for your positive feedback and for recognizing our contributions.
>
> -----------------
>
> Weaknesses 1: The primary concern regarding this paper is that the established minimax optimality for kernel classifiers relies on the gradient flow algorithm, which is mainly based on the L2 loss and is not commonly used in practical applications of building kernel classifiers. While the minimax rate is established, its optimality is only proven in an asymptotic sense, leaving a considerable gap between theory and practical usage. This approach is natural in the existing work when building the minimax rate for kernel classification, but not for the kernel classification. The paper's contribution would be more significant if the theoretical framework were based on widely used classifiers, such as SVM and logistic regression. A similar issue arises with the application of kernel classifier theory to neural networks, particularly those employing the L2 loss for fitting. Despite the L2 loss being effective in various neural network scenarios, the paper would benefit from incorporating theories that utilize loss functions more prevalent in practice.
>
> Answer: In respect of loss function, we have the following considerations. For SVM (hinge loss), more works about RKHS classification such as [1,2] have been studied, but they are far from recent classification works based on gradient algorithms. Logistic regression or cross-entropy loss is more popular in recent works, but in fact, there is no clear evidence that cross-entropy loss can work better than the squared loss function. In contrast, analyzing the classification upper bound of logistic regression in statistics [3] requires adding some additional unnatural conditions. Cross-entropy loss is related to maximum likelihood estimate (MLE) and high-dimensional statistics use MLE because it is a convex problem, which is not needed for our work.
>
> For the minimax rate, it is related to the worst case and seems far from practice, but it is meaningful. For example, we obtained the minimax optimal rate by choosing a proper $t \asymp n^{\frac{ \beta }{  s\beta + 1}}$ in Theorem 2 and Corollary 1, which can guide us to find early stopping criteria in practice \cite{raskutti2014early}. But we admit that we should connect more to practice in the further work.
>
> -------------
>
> Weakness 2: An additional point of concern is that the paper's main theoretical underpinnings are based on unpublished works, specifically referenced as https://arxiv.org/abs/2305.07241. The validity and proofs presented in that work have not been verified in this review.
>
> Answer: Thanks for reminding us. We have checked the proofs of this paper and it has been accepted by ICML 2023 (https://proceedings.mlr.press/v202/zhang23x.html)
>
> -------------
> Question: After estimating the smoothing parameter, can we further gain some insight into the optimal steps in the gradient flow algorithms? It would be helpful if some numerical studies can be performed to support this.
>
> Answer: It is easy to get that if $t$ is too large or too small, the estimator can overfit or underfit the data. The main idea of the optimal steps is to balance the bias (the approximation error) and the variance (the estimation error). The upper bound of the classification excess risk can be decomposed by the bias term $||f_t - f^*_\rho||_{L^2} = O(t^{-s/2})$, where $f_t$ can be considered as the estimator with infinite data points, and the variance term $||\hat{f}_t- f_t||_2=O\left(\frac{t^{\frac{1}{2\beta}}}{\sqrt{n}}\right)$. Thus, the optimal steps $t\asymp n^{\beta/(s\beta+1)}$.
>
> For example, we use the same classification settings in Section 5: $f^*(x)=\cos(2\pi x)$, $\mathbb{P}(Y=1|X=x)=(f^*(x)+1)/2$ and $K_{min}(x,x') = \min(x,x')$ is considered. Thus, the optimal $t=n^{\frac{2}{0.5\times 2 +1 }}=n$. Denote $t=n^{\theta}$ and we choose different $\theta=0.2,0.6,1,2,3$ to see the bias term, the variance term and the accuracy.
>
> $\theta$ $\mid$ 0.2  $\mid$ 0.6  $\mid$  $\textbf{1}$    $\mid$ 2    $\mid$ 3
>
> Bias     $\mid$ 0.65 $\mid$ 0.30 $\mid$ $\textbf{0.11}$ $\mid$ 0.01 $\mid$ 0
>
> Var      $\mid$  0.04 $\mid$  0.06 $\mid$  $\textbf{0.09}$ $\mid$  0.43 $\mid$  0.70
>
> Acc      $\mid$  0.83 $\mid$  0.94 $\mid$ $\textbf{0.97}$ $\mid$  0.82 $\mid$ 0.82
>
>
>
>
>
>
>
> ------------
>
> Reference:
>
> [1]  Ingo Steinwart and Andreas Christmann. Support vector machines. Springer Science & Business Media, 2008.
>
> [2] Gilles Blanchard, Olivier Bousquet, and Pascal Massart. Statistical performance of support vector machines. The Annals of Statistics, 36(2):489–531, 2008.
>
> [3] Felix Abramovich and Vadim Grinshtein. High-dimensional classification by sparse logistic regression. IEEE Transactions on Information Theory, 65(5):3068–3079, 2018.

---

### Official Review · Reviewer_ABCa · 2023-11-10

**Soundness:** 3 good
**Presentation:** 3 good
**Contribution:** 2 fair
**Rating:** 6
**Confidence:** 2

**Summary:**

This paper considers estimating classifiers in the interpolation Sobolev RKHS. They have two theoretical contributions: (a) the lower bound on the excess risk and (b) the upper bound of the same at the t-th iterate. They apply this results to a neural network which can be approximated by a neural tangent kernel. The optimal rate depends on the smoothness of the optimal classifier, larger smoothness leading to faster convergence. They propose a truncation strategy for estimation of the smoothness, which when applied to real datasets corraborates the estimated smoothness with the difficulty of the dataset.

**Strengths:**

The paper is well written and easy to follow. The main results seem novel for the setup considered. The association of the estimated smoothness to the difficulty of the datasets is interesting.

**Weaknesses:**

see questions.

**Questions:**

-- Not very clear how the arrived results compare against the existing work on NTK. Would be interesting to understand that to appreciate the impact of the contributions
-- It is not very clear about the impact of this work from an application front. The experiments to real datasets are restricted to estimation of the smoothness. Some insights would be helpful in this regard. How to enforce learning estimators constrained with a requirement on the smoothness.

---

> ### Author Response · Authors · 2023-11-15
> **Thank you for your feedback and questions.**
>
> Thank you for your feedback and questions. Your last question is so insightful and thought-provoking. We try our best to answer all your questions and the answers will be collected into our paper whether accepted or not.
>
> ---------------
>
> Question 1: Not very clear how the arrived results compare against the existing work on NTK. Would be interesting to understand that to appreciate the impact of the contributions.
>
> Answer: The most relevant result is Theorem 3.1 in [1], showing that if $f\in H_{ntk}$, then the upper bound of the generalization error rate is $n^{-\frac{\beta}{2(\beta+1)}}$ ($\kappa=0$ for our case in [1]).  However, we emphasize that $[H_{ntk}]^1=H_{ntk}$ and the assumption $f\in H_{ntk}$ is too strong. The experiments in Section 5 of our paper show that this assumption fails on the real data ($s<1$). However, our assumption $f\in[H_{ntk}]^s$  is so weak that even for small $s$, we show the upper bound of the generalization error $n^{-\frac{s\beta}{2(s\beta+1)}}$ and it matches the minimax lower bound. Our results are more general and more practical since the smoothness of the real data can be estimated (Section 5).
>
> -----------------
>
> Question 2: It is not very clear about the impact of this work from an application front.
>
> Answer: As mentioned before, we can estimate the smoothness of the real data with respect to the kernel. That means that given the real data, we can do the kernel selection for the real data with larger smoothness. For example, [2] showed that the decay rate of NTK with any depth is equal to $\beta=d/(d-1)$. Then we can test the smoothness of the real data with respect to NTKs with different depths and then find the best NTK. For the kernels with different $\beta$, we can try to find the kernel with the highest $s\beta$.
>
> ---------------
>
> Question 3: The experiments with real datasets are restricted to the estimation of the smoothness. Some insights would be helpful in this regard. How to enforce learning estimators constrained with a requirement on the smoothness.
>
> Answer: We can first consider the extreme case $s=\infty$, meaning that the ground true function  $f^*(x)=\sum_{j=1}^{J}f_j e_j(x)$, where $J$ is finite. In this case, the problem becomes a linear case ($\{f_j\}$ are the coefficients and $\{e_j(x)\}$ are the independent variables). Thus, the optimal rate is $n^{-\frac{1}{2}}$.
>
> For the case with finite $s$, the smoothness $s$ presents the alignment of $f^*$ with the top eigenfunctions. The larger $s$, the more projection of $f^*$ is on the top eigenfunctions, meaning that the problem is more 'linear'(like the linear case), and thus the estimator can more easily learn the ground true function $f^*$.
>
> -----------
>
> Reference:
>
> [1]  Tianyang Hu, Jun Wang, Wenjia Wang, and Zhenguo Li. Understanding square loss in training overparametrized neural network classifiers. arXiv preprint arXiv:2112.03657,2021.
>
> [2] Alberto Bietti and Francis Bach. Deep equals shallow for relu networks in kernel regimes. arXiv preprint arXiv:2009.14397, 2020.

---

> > ### Comment · Reviewer_ABCa · 2023-11-23
> > **Thanks for the clarifications**
> >
> > I thank the authors for the clarifications.

---

### Meta-Review · Area_Chair_Bf4w · 2023-12-05

**Metareview:**

This is a borderline paper. After significant discussions between reviewers and authors the consensus is that it is worth publishing the paper despite some criticism and shortcomings. It would be useful if the authors could address in the final version the detailed reviewer comments and potentially some of the bigger points raised by the reviewers.

**Justification For Why Not Higher Score:**

The reviewer scores are around 6 and there are a number of shortcomings in this paper which prevent me from suggesting a higher score.

**Justification For Why Not Lower Score:**

This is borderline paper and a reject was not out of the question in the discussions.

---

### Decision · Program_Chairs · 2024-01-16

Accept (poster)